# Visualizing synaptic plasticity in vivo by large-scale imaging of endogenous AMPA receptors

Austin R Graves[1,2†], Richard H Roth[1†], Han L Tan[1†], Qianwen Zhu[1†], Alexei M Bygrave[1†], Elena Lopez-Ortega[1†], Ingie Hong[1], Alina C Spiegel[1,2], Richard C Johnson[1], Joshua T Vogelstein[2,3,4], Daniel J Tward[2,3,4], Michael I Miller[2,3,4], Richard L Huganir[1,2]*

[1]Department of Neuroscience, Johns Hopkins University School of Medicine, Baltimore, United States; [2]Kavli Neuroscience Discovery Institute, Baltimore, United States; [3]Center for Imaging Science, Johns Hopkins University School of Engineering, Baltimore, United States; [4]Department of Biomedical Engineering, Johns Hopkins University, Baltimore, United States

**Abstract** Elucidating how synaptic molecules such as AMPA receptors mediate neuronal communication and tracking their dynamic expression during behavior is crucial to understand cognition and disease, but current technological barriers preclude large-scale exploration of molecular dynamics in vivo. We have developed a suite of innovative methodologies that break through these barriers: a new knockin mouse line with fluorescently tagged endogenous AMPA receptors, two-photon imaging of hundreds of thousands of labeled synapses in behaving mice, and computer vision-based automatic synapse detection. Using these tools, we can longitudinally track how the strength of populations of synapses changes during behavior. We used this approach to generate an unprecedentedly detailed spatiotemporal map of synapses undergoing changes in strength following sensory experience. More generally, these tools can be used as an optical probe capable of measuring functional synapse strength across entire brain areas during any behavioral paradigm, describing complex system-wide changes with molecular precision.

*For correspondence:
rhuganir@jhmi.edu

†These authors contributed equally to this work

## Introduction

Recent applications of genetically encoded calcium indicators and high-density silicon electrodes have revolutionized our understanding of the cellular and circuit basis of behavior; however, technological barriers preclude similar exploration of the molecular basis of these processes in vivo. To investigate the physiological function of complex molecular systems in vivo, we require techniques to visualize endogenous proteins. Modern proteomic and transcriptomic methods provide biologists with myriad candidate proteins, but in many cases, there are no tools available to effectively study these targets at the level of endogenous proteins in vivo. For example, we are far from having reliable antibodies for the entire proteome, and even when antibodies are available, there are concerns regarding their target specificity. Another approach is to fluorescently tag proteins to visualize their dynamic expression in living tissue. Combined with in vivo two-photon (2p) microscopy, this approach enables detailed investigation of the molecular mechanisms underlying complex physiological and pathological systems.

AMPA-type glutamate receptors (AMPARs) are crucial molecules to study to understand the function and dynamics of the nervous system. AMPARs mediate the majority of fast excitatory synaptic transmission in the mammalian brain, and their regulation is regarded as a key mechanism underlying

long-lasting changes in synaptic efficacy that give rise to learning and memory (*Huganir and Nicoll, 2013*; *Malinow and Malenka, 2002*). Long-term potentiation (LTP) is characterized by increased AMPAR trafficking to the postsynaptic membrane and associated spine enlargement, which together result in a long-lasting increase in synaptic efficacy, whereas long-term depression (LTD) is characterized by removal of postsynaptic AMPARs, resulting in attenuated synaptic transmission (*Anggono and Huganir, 2012*; *Nicoll, 2017*). Impaired regulation of synaptic plasticity is associated with human neurological and psychiatric disease (*Berryer et al., 2013*; *Henley and Wilkinson, 2016*; *Volk et al., 2015*). Despite this clear link between synaptic plasticity and learning, as well as a thorough understanding of molecular mechanisms regulating AMPAR trafficking, very little is known regarding how changes in plasticity are distributed among trillions of synapses throughout the brain. In contrast to advanced strategies to observe and manipulate neuronal activity – using genetically encoded calcium indicators (*Dombeck et al., 2010*; *Lin and Schnitzer, 2016*; *Xu et al., 2012*) or optogenetics (*Fenno et al., 2011*), respectively – there are currently no methods to physiologically measure postsynaptic strength in vivo on a brain-wide scale.

To overcome this barrier, we developed a new knockin mouse line wherein the AMPAR GluA1 subunit is tagged with super ecliptic pHluorin (SEP), a pH-sensitive variant of GFP that fluoresces at neutral pH and is quenched at acidic pH (*Miesenböck et al., 1998*). When coupled to the extracellular N-terminal domain of the AMPAR, this SEP tag reports the concentration of functional receptors at the cell surface, as the fluorescence of receptors localized in acidic, internal compartments such as endosomes and Golgi is quenched. Our genetic labeling strategy also avoids confounds arising from manipulation of the AMPAR C-terminus, a region important for proper function and trafficking to the postsynaptic membrane (*Sheng et al., 2018*; *Zhou et al., 2018*). Many groups have used overexpression of SEP-tagged AMPARs in neuronal culture to study AMPAR trafficking in vitro (*Araki et al., 2015*; *Ashby et al., 2004*; *Kopec et al., 2006*; *Makino and Malinow, 2009*; *Patterson et al., 2010*; *Roth et al., 2017*). In addition, previous work using overexpression of SEP-tagged AMPARs in vivo has provided valuable insights regarding the molecular mechanisms of behaviorally relevant plasticity (*Diering et al., 2017*; *El-Boustani et al., 2018*; *Makino and Malinow, 2009*; *Miyamoto et al., 2021*; *Roth et al., 2020*; *Suresh and Dunaevsky, 2017*; *Tan et al., 2020*; *Zhang et al., 2015*), but these methods enable receptor visualization in only a sparse subset of cells and exogenous overexpression may result in protein mistargeting and dysregulation. The novel genetic labeling strategy presented here avoids these confounds, allowing visualization of endogenous AMPAR expression in a manner that does not impair synaptic function, plasticity, or behavior. Used in conjunction with in vivo 2p microscopy, this novel SEP-GluA1 knockin mouse is the first tool that enables longitudinal tracking of synaptic plasticity underlying behavior at brain-wide scale with single-synapse resolution. Finally, we present a suite of algorithms to automatically detect and segment hundreds of thousands of fluorescently labeled AMPARs in vivo, enabling longitudinal tracking of synaptic plasticity across entire brain regions in awake behaving mice.

## Results
### SEP-GluA1 knockin mouse line labels excitatory synapses

Using homologous recombination, we generated a mouse knockin line (C57BL/6J background) that inserts SEP into the N-terminus of *GRIA1*, the gene that encodes the GluA1 AMPAR subunit. Homozygous knockin mice are viable, breed well, and appear to be physiologically and behaviorally normal (see below). This approach fluorescently labels all GluA1-containing AMPARs in the mice (*Figure 1a–c*), enabling robust visualization of excitatory synapses throughout the entire brain (*Figure 1—figure supplement 1*). We did observe a decrease in GluA1 mRNA and protein expression in our homozygote knockin line compared to wild type (WT; *Figure 1d–f*), most likely due to decreased stability of the resulting mRNA. Using biochemical fractionation to isolate synapses from mouse hippocampal tissue, we observed reduced expression of GluA1 in the postsynaptic density (PSD, 56.6% of WT mice) and in total membrane protein levels (P2, 44.4% of WT mice; *Figure 1e and f*). In contrast, we observed a trend of increased levels of GluA2 (P2, 107.9% of WT mice; PSD, 115.8% of WT mice) and GluA3 (P2, 113.4% of WT mice; PSD, 125.6% of WT mice) subunits in knockin mice, although these changes were not significant (*Figure 1e and f*). These results suggest that there might be a small compensatory increase of GluA2/GluA3 in the knockin line as a result of decreased GluA1 expression.

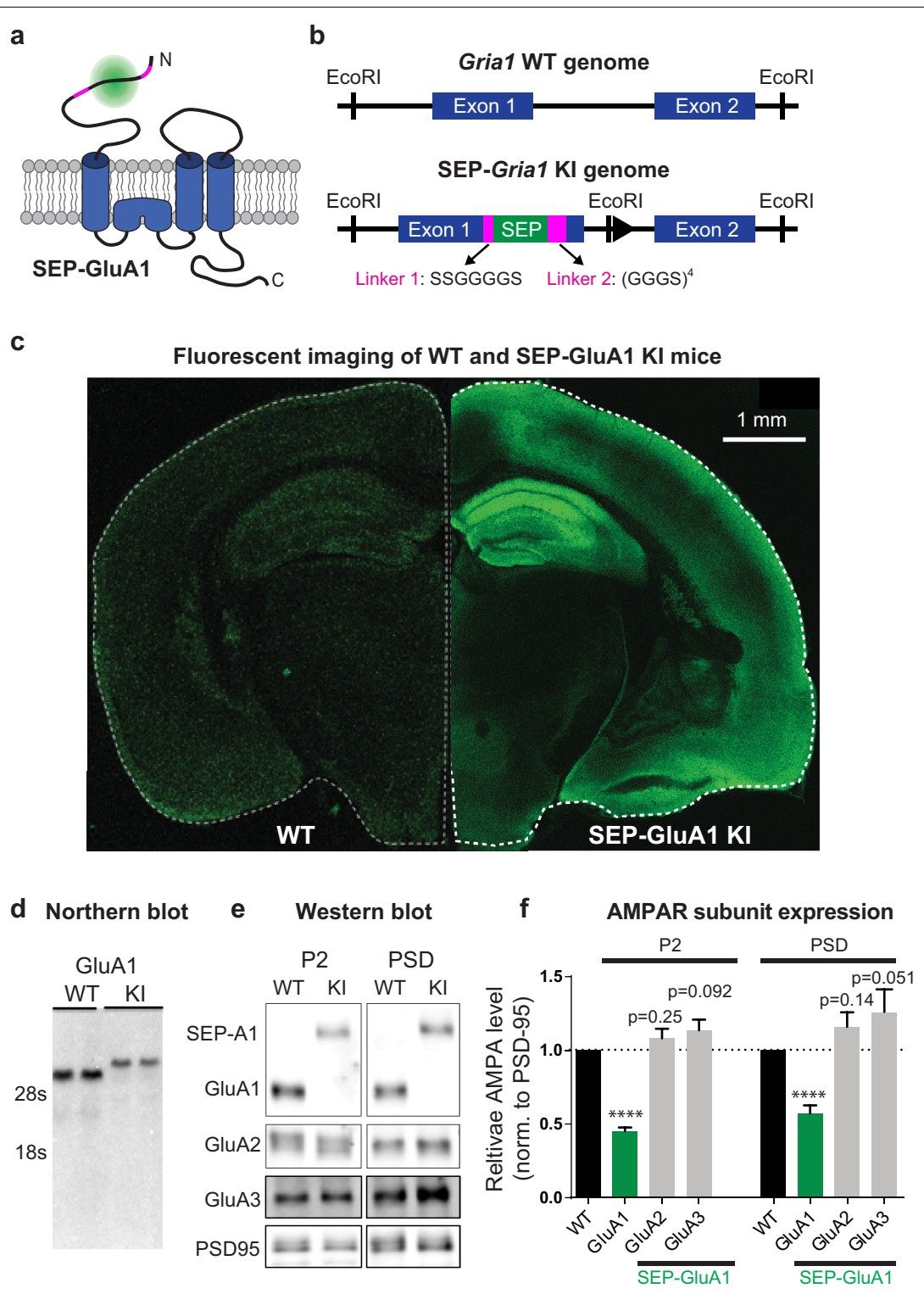

**Figure 1.** Generation and biochemical characterization of SEP-GluA1 knockin (KI) mouse line. (**a**) SEP tag (green) was targeted to extracellular N-terminus of GluA1 AMPARs (blue), enabling visualization of only the functional complement of AMPARs on the cell surface. SEP-GluA1 linkers depicted in magenta. (**b**) Schematic of genetic locus of SEP tag on exon 1, within the *Gria1* gene encoding GluA1. Two linkers flank the SEP insert. (**c**) Live, confocal image of acute slice of SEP-GluA1. Note the bright fluorescent signal throughout hippocampus and neocortex, indicating widespread expression of SEP-labeled GluA1-containing AMPARs. Age- and coronal region-matched wild-type (WT) tissue was imaged with the same laser power and presented with identical contrast as SEP-GluA1. (**d**) Representative northern blot of mRNA expression of WT and SEP-GluA1 KI mice. SEP-GluA1

*Figure 1 continued on next page*

*Figure 1 continued*

is noticeably larger than WT GluA1 due to the inclusion of the SEP tag. (**e, f**) Representative western blot and quantification of AMPA receptor subunit expression in hippocampus after normalization to PSD95 in the P2 and postsynaptic density (PSD) fractions of WT and SEP-GluA1 mice. GluA1 expression is reduced relative to WT (n = 7; ****p<0.0001, Student's t-test). Full, uncropped blots of all bands can be found in *Figure 1—source data 1*.

The online version of this article includes the following source data and figure supplement(s) for figure 1:

**Source data 1.** Uncropped images of western blots used to display representative bands in *Figure 1e*.

**Source data 2.** Original files of western blots used to display representative bands stained for GluA1 and GluA3 in *Figure 1e*.

**Source data 3.** Original files of western blots used to display representative bands stained for GluA2 and PSD-95 in *Figure 1e*.

**Source data 4.** Quantification of Western blots.

**Figure supplement 1.** Expression atlases of SEP-GluA1 and *Gria1*.

**Figure supplement 2.** Region-specific differences in GluA1 expression in SEP-GluA1 KI mice.

Western blots of total lysates from either whole-brain or individual regions (hippocampus, cortex, and cerebellum) revealed that GluA1 expression levels were consistent across brain regions in knockin mice (*Figure 1—figure supplement 2*).

To validate the physiological function of our knockin line, we made whole-cell voltage-clamp recordings from CA1 pyramidal neurons in acute hippocampal slices of WT and homozygous SEP-GluA1 littermates. We observed no deficits in synaptic physiology or receptor trafficking in SEP-GluA1 mice (*Figure 2*). Our electrophysiological data in particular support that synapses with fluorescently labeled AMPARs function identically to WT synapses, with no discernible differences in the amplitude, frequency, or kinetics of miniature excitatory postsynaptic currents (mini EPSCs; *Figure 2a–c*) or rectification of the EPSCs (*Figure 2d and e*). We also observed that SEP-tagged GluA1 receptors were properly trafficked to the postsynaptic site and colocalized normally with other postsynaptic proteins, such as PSD-95 (*Figure 2f–h*). There was a highly significant correlation between SEP-GluA1 signal and immunofluorescence intensity of both PSD-95 and c-terminal GluA1 antibodies, indicating that all GluA1 receptors express SEP in our knockin mouse line (*Figure 2g–i*). Small differences in the appearance of dendritic shaft staining between SEP-GluA1 staining and c-terminal GluA1 staining might reflect a degree of nonspecific binding of the c-terminal GluA1 antibody or differences in epitope accessibility that result in higher apparent dendritic signal with c-terminal GluA1 staining (*Figure 2f*). However, the clear overlap of SEP-GluA1 and c-terminal GluA1 at synaptic puncta (*Figure 2g and i*) suggests proper trafficking and expression of SEP-GluA1.

## Intact synaptic plasticity and normal behavior in SEP-GluA1 knockin mice

Homeostatic and Hebbian plasticity are the two major forms of synaptic plasticity that function cooperatively to keep neural circuits stable and plastic, respectively (*Bliss and Lomo, 1973*; *O'Brien et al., 1998*; *Turrigiano et al., 1998*). This novel knockin line represents a powerful tool to study both of these mechanisms as they are each known to be expressed via dynamic regulation of synaptic AMPARs. To evaluate if homeostatic plasticity is intact in our knockin line, we made primary cultures of cortical neurons from homozygous SEP-GluA1 mice and WT littermates and treated them with either tetrodotoxin (TTX) or bicuculline for 2 days to induce up- or downscaling, respectively (*Figure 3a and b*). As expected, WT neurons exhibited a significant reduction of surface AMPARs following bicuculline treatment and showed an elevation of surface GluA1 and GluA2 after TTX treatment. SEP-GluA1 knockin neurons displayed a similar bidirectional change of surface AMPARs following bicuculline and TTX treatments, indicating comparable homeostatic plasticity in SEP-GluA1 and WT mice. To assess Hebbian plasticity, we compared the expression of LTP in WT and homozygous SEP-GluA1 littermates (*Figure 3c–e*) as GluA1 knockout mice show deficits in LTP (*Zamanillo et al., 1999*). We performed whole-cell voltage-clamp recordings of synaptically evoked EPSCs in CA1 pyramidal cells in acute hippocampal slices of 3–4-week-old mice. After a baseline period of at least 5 min, a pairing stimulus consisting of 2 Hz synaptic stimulation and somatic depolarization to 0 mV was delivered, after which we resumed monitoring the amplitude of evoked EPSCs (*Figure 3d*). This pairing protocol induced a long-lasting increase in EPSC amplitude in both WT and knockin neurons (n = 8 cells from each genotype), consistent with induction of LTP. We observed no differences in either induction or expression

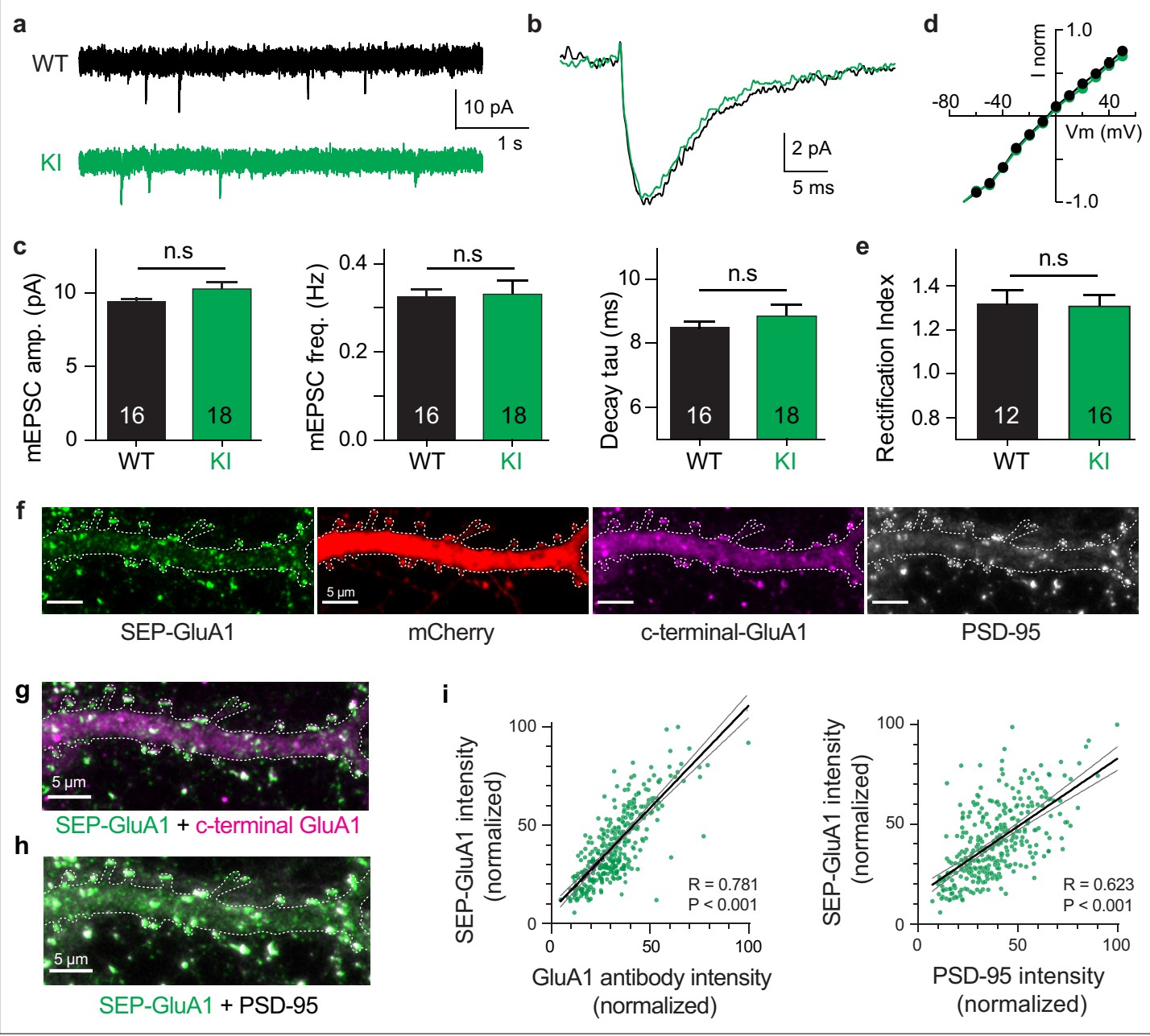

**Figure 2.** Normal synaptic physiology and receptor trafficking in SEP-GluA1 knockin (KI) mice. (**a–e**) Whole-cell voltage-clamp recordings from CA1 neurons in acute hippocampal slices of aged-matched wild-type (WT) and homozygous SEP-GluA1 KI littermates. (**a, b**) Representative traces of miniature excitatory postsynaptic currents (EPSCs) from WT (black) and SEP-GluA1 knockinKI (green) mice. (**c**) Quantification of amplitude, frequency, and kinetics of miniature EPSCs. No differences were observed in any electrophysiological parameters between WT and KI mice. N = 16 and 18 cells from WT and KI, respectively. Unpaired t-tests were used for all comparisons. Mean ± SEM. mEPSC amplitude: WT 9.43 ± 0.20, N = 16; KI 10.26 ± 0.51, N = 18; p>0.05. Frequency: WT 0.33 ± 0.02; KI 0.33 ± 0.03122; p>0.05. Rise time: WT 2.46 ± 0.040; KI 2.49 ± 0.07; p>0.05. Tau decay: WT 8.49 ± 0.18; KI 8.83 ± 0.36; p>0.05. (**d, e**) No differences in rectification were observed between WT and KI mice. Rectification index is the negative slope of the IV curve (between –10 and –60 mV) divided by the positive slope (between +10 and + 50 mV). Mean ± SEM. WT: 1.32 ± 0.048, N = 12; KI: 1.33 ± 0.042, N = 16; p>0.05. (**f–h**) Confocal images of cultured SEP-GluA1 (green) neurons, with an mCherry cell-fill (red) and stained with antibodies for c-terminal-GluA1 (magenta) and PSD-95 (gray). Overlap of SEP-GluA1 with c-terminal-GluA1 (**g**) and SEP-GluA1 with PSD-95 (**h**) is rendered in white. (**i**) Quantifying overlap between endogenous SEP-GluA1 signal and immunofluorescence. A significant correlation is observed between the fluorescent intensity of endogenous SEP and the immunofluorescent signal of both GluA1 (R = 0.781, p<0.001, Pearson correlation, n = 332 spines) and PSD-95 (R = 0.623, p<0.001, Pearson correlation, n = 332 spines). Lines represent linear regression (thick black) with 95% confidence interval (thin gray).

The online version of this article includes the following source data for figure 2:

**Source data 1.** Raw data used to make *Figure 2* plots.

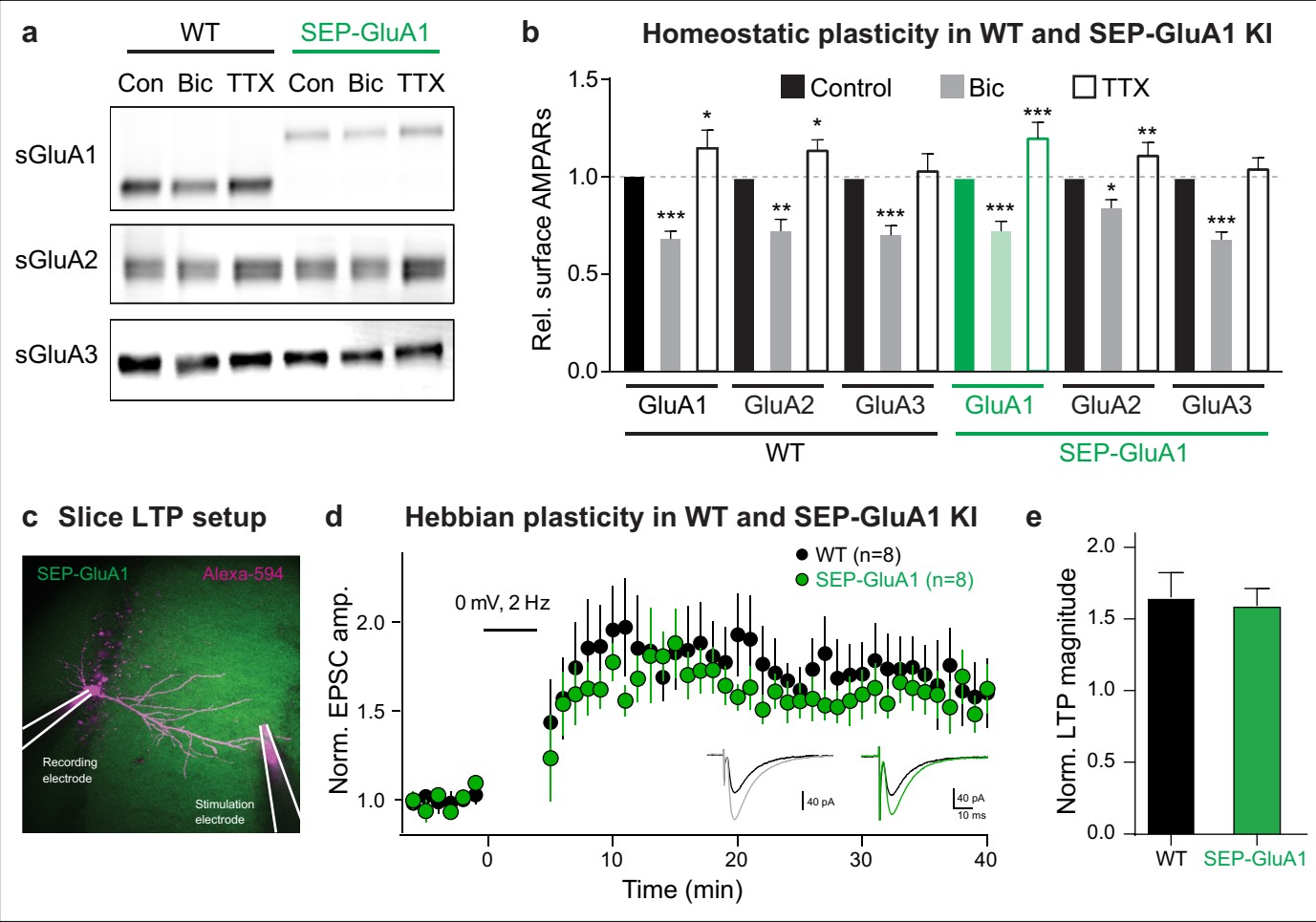

**Figure 3.** Normal homeostatic and Hebbian plasticity in SEP-GluA1 knockin (KI) mice. (**a**) Representative western blot of surface GluA1, GluA2, and GluA3 in wild-type (WT) and KI mouse neurons under baseline conditions (Con), following homeostatic downscaling in bicuculline (Bic), and following homeostatic upscaling in tetrodotoxin (TTX). (**b**) Bar plot of all homeostatic plasticity experiments (n = 7–8; *p<0.05, **p<0.01, ***p<0.001; one-way ANOVA). (**c**) Induction of long-term potentiation (LTP) in WT and SEP-GluA1 KI mice. Fluorescent image depicting experimental setup. CA1 pyramidal neurons were patched and filled with Alexa-594. A stimulating electrode in *stratum radiatum* was used to evoke excitatory postsynaptic currents (EPSCs). After recording baseline EPSCs for at least 5 min, a pairing protocol consisting of 200 pulses was delivered at 2 Hz. (**d**) Average EPSC amplitude normalized to baseline for WT (black) and KI (green) littermates over course of LTP induction. Inset: example traces of EPSCs from baseline (black) and 30–40 min following LTP induction (WT gray and KI green). (**e**) Average change in EPSC amplitude normalized to the baseline period for WT and SEP-GluA1 KI littermates. A significant potentiation of EPSC amplitude was observed in both WT and KI mice, which was not different between genotypes, indicating normal induction and expression of LTP in SEP-GluA1 KI mice. Unpaired t-tests, p>0.05, mean ± SEM; WT: 1.66 ± 0.17, n = 8; KI: 1.59 ± 0.34, n = 8. Full, uncropped blots of all bands can be found in *Figure 1—source data 1*.

The online version of this article includes the following source data and figure supplement(s) for figure 3:

**Source data 1.** Uncropped images of western blots used to display representative bands in *Figure 3a*.

**Source data 2.** Original files of western blots used to quantify surface expression of GluA1 in *Figure 3a*.

**Source data 3.** Original files of western blots used to quantify surface expression of GluA2 in *Figure 3a*.

**Source data 4.** Original files of western blots used to quantify surface expression of GluA3 in *Figure 3a*.

**Source data 5.** Raw data used to make *Figure 3* plots.

of LTP between WT and SEP-GluA1 littermates (*Figure 3e*). Overall, these data strongly support that our knockin labeling strategy does not impair synaptic transmission and plasticity.

To further validate our SEP-GluA1 knockin line, we conducted a battery of behavioral experiments as GluA1 knockout mice show deficits in several behaviors, including locomotor activity, anxiety, and spatial memory (*Bannerman et al., 2004*; *Boerner et al., 2017*; *Bygrave et al., 2019*; *Sanderson et al., 2007*). We assessed these behaviors in cohorts of SEP-GluA1 and WT littermates that were

age-matched (both 6–10 weeks) and contained similar numbers of both sexes (SEP-GluA1: nine females and seven males; WT: nine females and nine males). We assessed locomotor activity by placing animals in an open arena and measuring the number of beam breaks during a 30 min session. We observed no differences in the time course or total number of beam breaks between WT and SEP-GluA1 mice (*Figure 4a and b*). Anxiety was assessed using an elevated plus maze, consisting of two closed arms and two open arms, suspended above the ground. We observed no differences in time spent in the open arms between WT and SEP-GluA1 mice (*Figure 4c and d*). Spatial short-term memory was assessed using a Y-maze, consisting of three arms and surrounded by distal spatial cues. During the initial exposure phase, one arm was blocked with a clear plexiglass barrier. After exploring the two unblocked arms of the maze, mice were returned to their home cage for 1 min, and then re-exposed to the maze for the test phase, wherein the barrier was removed. WT and SEP-GluA1 mice displayed a similar preference for the novel arm (*Figure 4e and f*). Overall, these data strongly support that our knockin labeling strategy does not impair behavior as SEP-GluA1 mice display comparable locomotion, anxiety, and short-term memory to WT animals.

## SEP-GluA1 reports synaptic plasticity in vitro

To examine the function of individual SEP-GluA1 synapses, we used whole-cell voltage-clamp recordings from primary cultures of homozygous SEP-GluA1 pyramidal neurons to measure evoked responses with 2p glutamate uncaging. To visualize dendritic spines, neurons were filled with a red fluorescent dye via the patch pipette. Glutamate uncaging was targeted to the tip of spine heads (*Figure 5a*) and the resulting uncaging-evoked excitatory postsynaptic current (uEPSC) was recorded (*Figure 5b*). We found a significant correlation between SEP-GluA1 fluorescence intensity and uEPSC amplitude (*Figure 5c*), indicating that SEP fluorescent intensity can be used as a proxy for synaptic strength.

We further used the SEP-GluA1 knockin line to track changes in synaptic strength following induction of synaptic plasticity in vitro in primary cultures of hippocampal neurons. Using 2p imaging, glutamate uncaging, and whole-cell patch-clamp recordings from pyramidal neurons, we tracked SEP intensity and functional synaptic strength of spines that received high-frequency glutamate uncaging paired with postsynaptic depolarization versus spines that did not receive this pairing. We found that this pairing stimulus significantly increased uEPSC amplitude and SEP fluorescence in stimulated spines (n = 10; p<0.01 relative to baseline, one-way ANOVA), consistent with induction of LTP, whereas spines that received only postsynaptic depolarization unpaired with glutamate uncaging did not display similar changes (n = 42 spines; *Figure 5d–f*). The strong correlation between SEP-GluA1 intensity and uEPSC amplitude is observed in both potentiated and unpotentiated spines, suggesting that the concentration of synaptic GluA1-containing AMPARs reflects functional synapse strength at baseline and following plasticity (*Figure 5—figure supplement 1*). These data highlight the power of the SEP-GluA1 knockin line as a tool to monitor widespread synaptic strength and plasticity via fluorescence imaging.

## SEP-GluA1 expression is regulated by a dynamic process in vivo

To confirm that SEP-GluA1 fluorescence reports AMPAR dynamics in vivo, we used in vivo 2p fluorescence recovery after photobleaching (FRAP) to measure SEP-GluA1 turnover rate. To visualize dendritic spines, AAV-CaMKII-Cre viral injections were performed in L2/3 of somatosensory cortex in SEP-GluA1 knockin × Ai9 (a tdTomato reporter line) double homozygous mice. The same area of cortex was imaged at baseline and following photobleaching (*Figure 6a*). Spines targeted for photobleaching resulted in ~50% reduction of fluorescence in SEP-GluA1 and ~13% decrease in fluorescence in tdTomato (*Figure 6—figure supplement 1*). FRAP results confirmed that SEP-GluA1 signal in our knockin line represents slowly mobile molecules, supporting that SEP-AMPARs are normally targeted to the plasma membrane, in contrast to the tdTomato signal, which is freely diffusible and thus recovers more quickly (*Figure 6—figure supplement 1*). We also found that the SEP-GluA1 signal recovered in two phases after photobleaching (*Figure 6b*). In the initial exponential phase (up to 30 min), SEP-GluA1 reaches a plateau at 50% fluorescence recovery. This result suggests that about half of the GluA1-containing AMPARs at the spines are part of the mobile fraction and, therefore, readily available to be exchanged with AMPAR pools outside the spine. This mobile fraction has been previously characterized by several in vitro and in vivo studies, although with different timescales. The timeline for mobile spine AMPAR exchange in cultured neurons has ranged from 5 min to periods

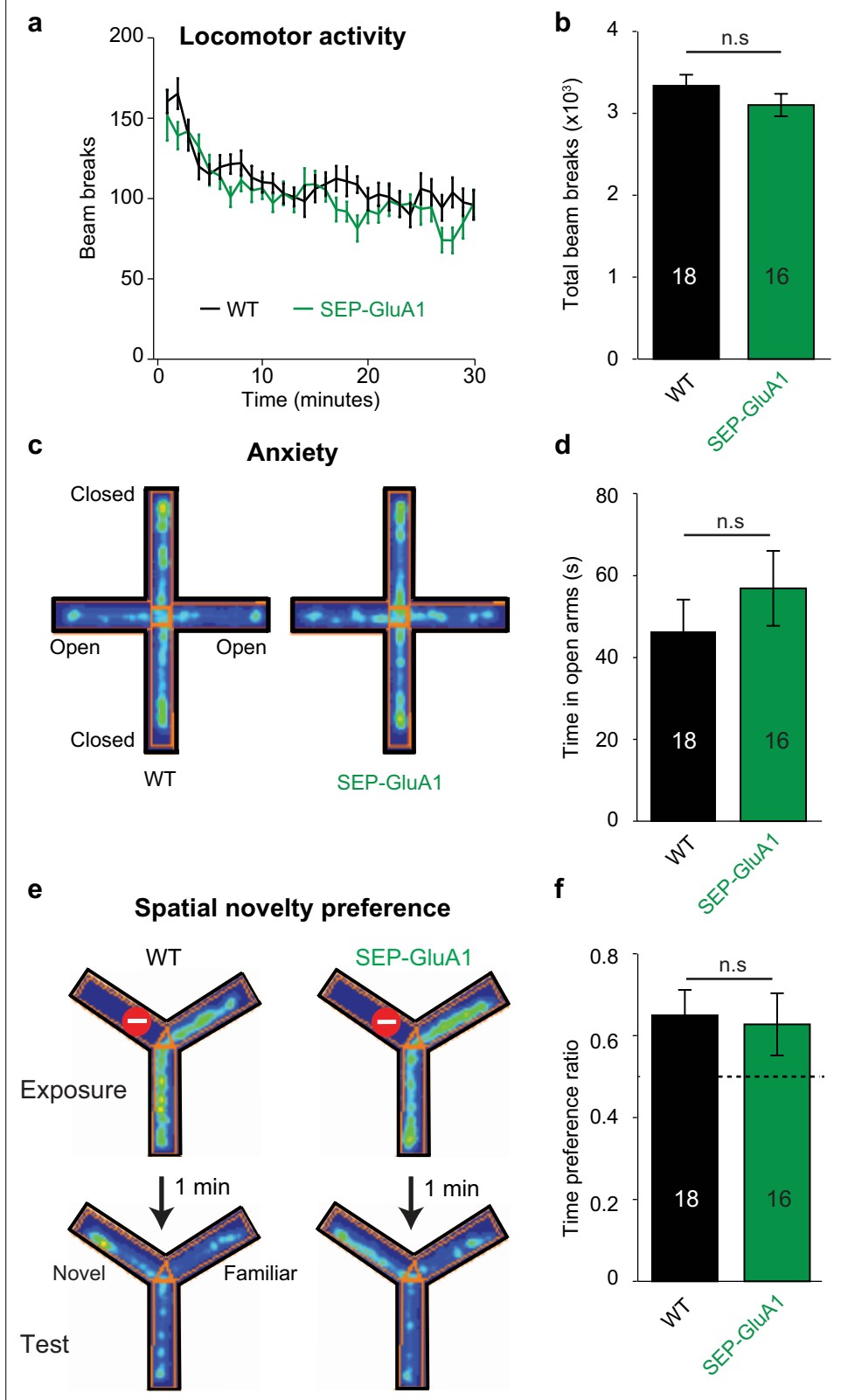

**Figure 4.** Normal behavior in SEP-GluA1 knockin (KI) mice. (**a, b**) SEP-GluA1 mice display normal locomotion. Age-matched, WT (n = 18) and homozygous SEP-GluA1 KI (n = 16) littermates were placed in an open chamber and locomotion was assessed by counting the total number of beam breaks in a 30 min session. No differences were detected between wild-type (WT) and KI mice (repeated measures ANOVA; $F_{1,30}$ = 1.561, p=0.221), between sexes

*Figure 4 continued on next page*

*Figure 4 continued*

($F_{1,30}$ = 1.346, p=0.255), or in a sex * genotype-dependent manner ($F_{1,30}$ = 0.3, p=0.588). (**c, d**) SEP-GluA1 mice display normal anxiety. WT (n = 18) and KI (n = 16) littermates were placed in an elevated plus maze and anxiety was assessed by measuring the time spent in the open arm. For representative WT and KI animals, time spent in a particular location is indicated in pseudo color, with warm colors indicating higher occupancy. No differences were detected between WT and KI mice (ANOVA; $F_{1,30}$ = 1.545, p=0.224), between sexes ($F_{1,30}$ = 0.160, p=0.692), or in a sex * genotype-dependent manner ($F_{1,30}$ = 4.139, p=0.051). (**e, f**) SEP-GluA1 mice display normal short-term spatial memory. Spatial novelty preference was assessed in WT (n = 18) and KI (n = 16) littermates using a Y-maze. WT and KI mice showed a preference for exploration of the novel arm. For representative WT and KI animals, time spent in a particular location is indicated in pseudo color, with warm colors indicating higher occupancy. There was no difference in the time preference ratio (time in novel arm/(time in novel arm + time in familiar arm)) between genotypes (ANOVA; $F_{1,30}$ = 0.004, p=0.951), between sexes ($F_{1,30}$ = 1.277, p=0.267), or in a sex * genotype-dependent manner ($F_{1,30}$ = 2.434, p=0.129). Dotted line indicates chance-level performance.

The online version of this article includes the following source data for figure 4:

**Source data 1.** Raw data used to make *Figure 4* plots.

longer than 15 min depending on the experimental conditions (*Ashby et al., 2006*; *Fang et al., 2021*; *Frischknecht et al., 2009*; *Lee et al., 2017*; *Martin et al., 2009*; *Sharma et al., 2006*). However, in the intact brain, the mobile AMPARs are exchanged after 30 min (*Chen et al., 2021*), in agreement with our observations. These results confirm that SEP-GluA1 is regulated by a cellular process that controls the dynamic exchange of molecules at the synapses, as would be expected for endogenous synaptic proteins.

The second phase at later time points represents a full recovery of SEP-GluA1 signal, indicating a complete turnover of synaptic GluA1-containing AMPARs in vivo within hours after photobleaching. This second AMPAR fraction is characterized by slower dynamics, probably due to protein interactions and molecular crowding within the PSD that limit AMPAR mobility (*Bats et al., 2007*; *Li et al., 2016*). The timescale presented in this study is supported by previous observations that establish a similar time course between synaptic AMPAR remodeling and AMPAR metabolic half-life (~18 hr for GluA1 subunit) in neuronal cultures (*Mammen et al., 1997*; *O'Brien et al., 1998*). In addition, these results demonstrate that we can track the same individual bleached and unbleached synapses longitudinally over several imaging sessions across consecutive days (*Figure 6a and c*). This is the first time that endogenous AMPAR recycling has been studied in vivo for up to 24 hr, providing valuable insights into basal AMPAR turnover dynamics in the intact brain and supporting that these tools can be used to longitudinally track synapse strength. Overall, these results confirm that SEP-GluA1-containing synapses are mobile and present similar dynamics as other in vitro and in vivo systems, supporting that our knockin labeling strategy does not perturb normal synaptic dynamics or function.

## Developing computational tools to detect and track labeled synapses in vivo

To observe AMPAR dynamics in living mice on a large scale, we implanted cranial windows over somatosensory cortex in homozygous SEP-GluA1 mice and used 2p microscopy to visualize endogenously labeled synapses (*Figure 7a–f* and *Videos 1–3*). The observed bright green punctate fluorescence reflects synaptic enrichment of GluA1, likely corresponding to the functional complement of GluA1-containing AMPARs at the PSD (*Figure 7b and c*). Given the richness and scale afforded by this knockin line (*Videos 1–3*), which endogenously labels all GluA1-containing synapses throughout the brain, manual annotation of labeled synapses was not feasible. Thus, to automatically detect and segment extremely large numbers of SEP-labeled synapses, we developed an unsupervised machine learning algorithm based on 3D Wiener filtering, employing pre-whitened matched templates based on the mean appearance of manually annotated synapses relative to background noise. This approach enabled flexibility to tune segmentations based on accuracy criteria, such as tradeoffs between sensitivity and specificity, as well as prior information about synapse size and shape.

To validate our computer vision-based automatic synapse detection platform, two expert synaptic anatomists manually annotated thousands of individual SEP-GluA1 synapses from in vivo volumes of somatosensory cortex (*Figure 7f*). We found relatively low inter-rater reliability (72.3% agreement, defined as >50% shared voxels; *Figure 7—figure supplement 1*). Thus, rather than attempting to

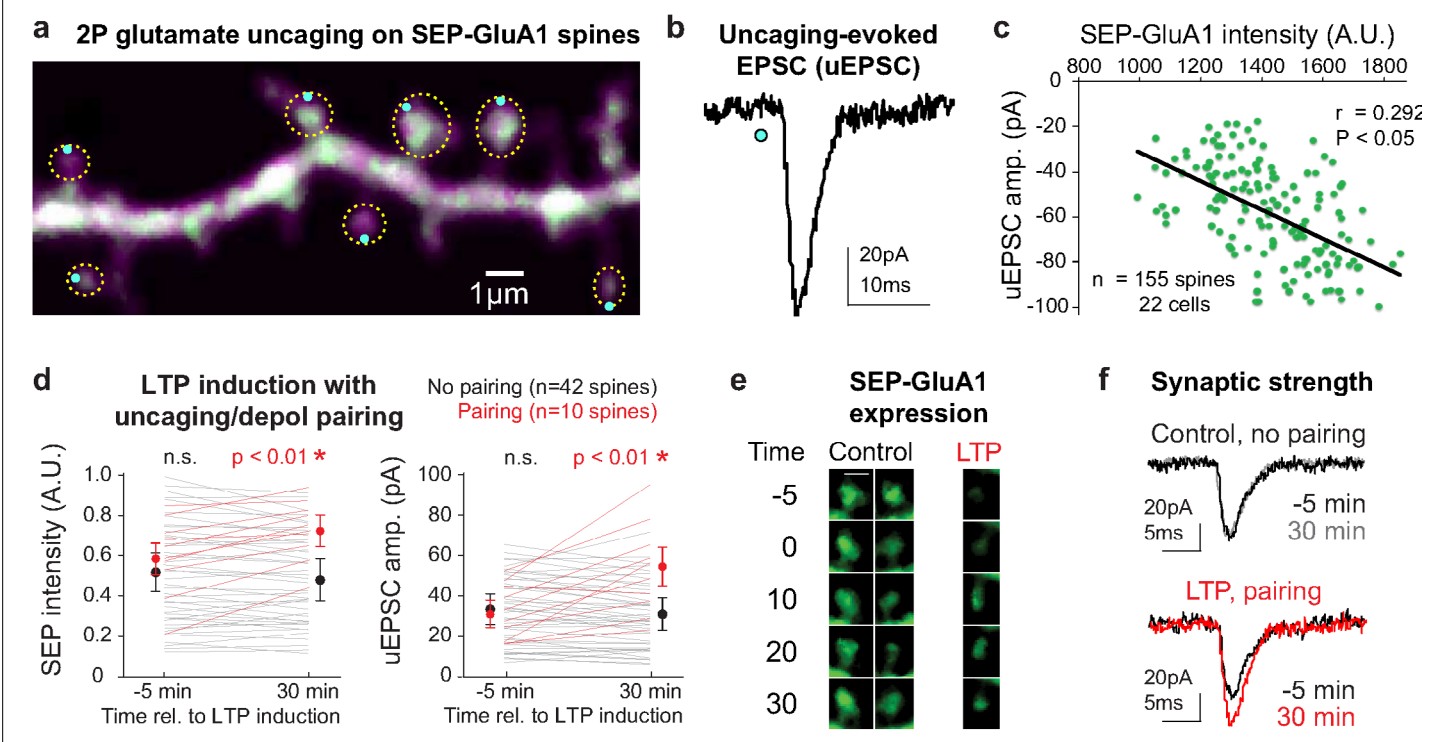

**Figure 5.** SEP intensity correlates with functional synaptic strength. (**a**) Two-photon (2p) image of a cultured SEP-GluA1 neuron filled with Alexa-594 via somatic patch pipette, imaged at 910 nm, with locations of glutamate uncaging indicated as blue dots and manually identified dendritic spines circled in yellow. (**b**) Representative uncaging-evoked excitatory postsynaptic current (uEPSC) following 1 ms pulse of 730 nm light at 20 mW (blue dot) in 2.5 mM MNI-glutamate. (**c**) Significant linear correlation between SEP-GluA1 intensity and uEPSC amplitude. SEP-GluA1 intensity was defined as the sum of green fluorescence intensity within manual synaptic annotations from five adjacent 0.5-μm-spaced Z-planes. n = 155 spines from 22 cells (*p<0.05, Pearson's chi-squared test). (**d–f**) Tracking synaptic plasticity with SEP-GluA1 in vitro. (**d**) Plots of SEP-GluA1 intensity and uEPSC amplitude 5 min before and 30 min after delivery of a long-term potentiation (LTP) induction stimulus consisting of high-frequency pairing of glutamate uncaging (30 pulses at 0.5 Hz, 1 ms pulse of 730 nm laser) and postsynaptic depolarization (0 mV for 0.5 s, beginning concurrently with uncaging pulse). Red, spines that received LTP stimulus (n = 10); black, spines that did not receive LTP stimulus (n = 42). *p<0.01 using one-way ANOVA. (**e**) Longitudinal images of two control spines and one spine that received LTP stimulus. (**f**) Representative uEPSCs during the baseline period (–5 min) and 30 min after LTP induction.

The online version of this article includes the following source data and figure supplement(s) for figure 5:

**Source data 1.** Raw data used to make *Figure 5* plots.

**Figure supplement 1.** Synaptic potentiation is expressed as increased SEP-GluA1 intensity and functional strength in vitro, and this correlation remains strong following long-term potentiation (LTP) in vitro.

**Figure supplement 1—source data 1.** Raw data used to make *Figure 5—figure supplement 1* plots.

design an algorithm that reproduces highly variable human intuition, we chose to carefully define what an observed synapse looks like through a system of rules. Our segmentation algorithm is unique in that we interpret its output as the physical definition of a synapse. Our algorithm uses the following rules to define a synapse and its boundaries: (1) a candidate synapse is defined as a local maximum in an image blurred using a Gaussian kernel with standard deviation of 5 pixels in the XY plane and 1 pixel out of plane; (2) candidate synapses are not less than 3 pixels away from each other, as determined by a furthest first traversal; (3) synapses are ellipses in the XY plane, with eccentricity between 1.0 and –2.5; (4) synapses have an area between 20 and 150 pixels in the XY plane, corresponding to a circle of area 0.125–1.25 μm² (though ovals were also considered); these constraints were based on mean synapse size from electron microscopy datasets (*Santuy et al., 2020*); (5) a synapse shape is chosen to be the size and orientation and eccentricity that maximizes signal-to-noise ratio (SNR) (template matching); (6) SNR should be larger than the 90th percentile of 300 randomly chosen locations; (7) averaging two neighboring slices should increase SNR; and (8) averaging six neighboring slices should decrease SNR.

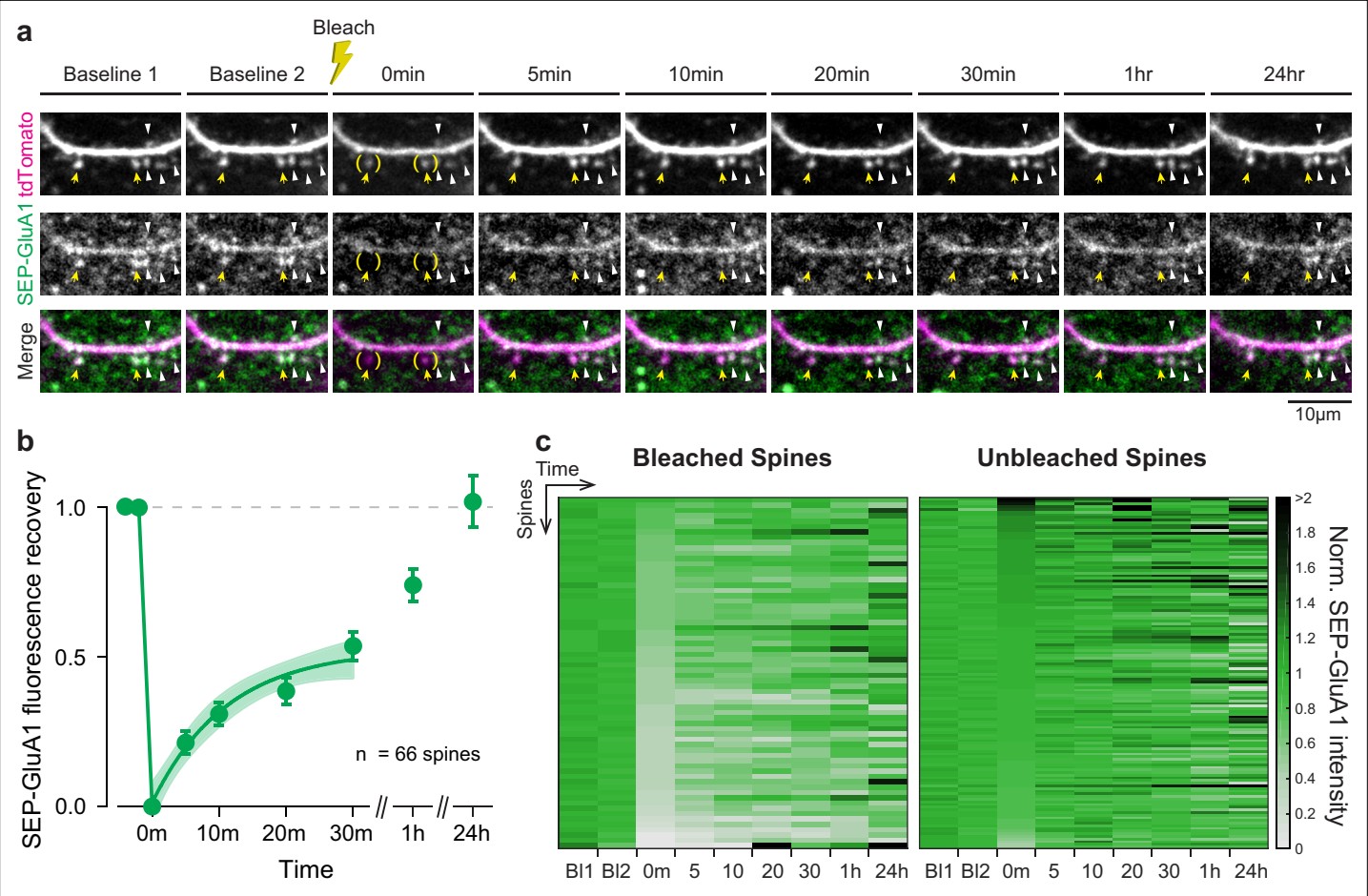

**Figure 6.** SEP-GluA1 signal completely recovers after photobleaching in vivo. (**a**) Representative in vivo two-photon (2p) images throughout fluorescence recovery after photobleaching (FRAP) in a SEP-GluA1 mouse. A sparse subset of neurons was filled with tdTomato to visualize dendrites and spines. Yellow arrows denote spines that were bleached (at t = 0 min; bleaching area depicted in parentheses) and spines that were not bleached are indicated by white arrows. Scale bar 10 μm. (**b**) Fluorescence recovery of SEP-GluA1 signal after photobleaching in spines of L2/3 excitatory neurons in mouse somatosensory cortex. Symbols represent mean and error bars represent SEM. Time points between 0 and 30 min were fitted to a one-phase decay exponential curve (solid line), with plateau = 0.526 ± 0.057, rate constant of recovery (k) = 0.09 ± 0.027 (value ± SEM) and tau = 11.15 min. Shaded area represents 95% confidence interval of the fit. n = 66 spines from three mice. (**c**) Heatmap of SEP-GluA1 signal from individual bleached and unbleached spines normalized to their respective baseline at different time points throughout FRAP. Rows represent individual spines sorted by signal intensity immediately after photobleaching (t = 0 min). Bleached spines: n = 66 spines from three mice. Unbleached spines: n = 132 spines from three mice.

The online version of this article includes the following source data and figure supplement(s) for figure 6:

**Source data 1.** Raw data used to make *Figure 6* plots.

**Figure supplement 1.** Fluorescence recovery of tdTomato cell fill and SEP-GluA1 signal after photobleaching.

By filtering based on size in both the XY and Z planes and by applying template matching, we were able to minimize false-positive detections, likely corresponding to either acquisition noise or extra-synaptic SEP-GluA1 receptors along the dendritic shaft. While fluorescent signal from extrasynaptic receptors is certainly present in our images (see *Videos 1–3*), our computational approach was largely successful in filtering out this smaller, more diffuse, and less punctate signal from our automated synapse detection and subsequent analyses.

To quantitatively assess ground truth for synapse detection, we performed immunohistochemical labeling of Homer, an abundant PSD protein, comparing rates of overlap between these two independent synaptic markers (*Figure 7—figure supplement 2*). As we currently do not have tools to label and visualize Homer expression in vivo, we imaged slices of barrel cortex from SEP-GluA1 mice using the same 2p beam path and identical acquisition settings as for in vivo experiments. SEP-GluA1 and

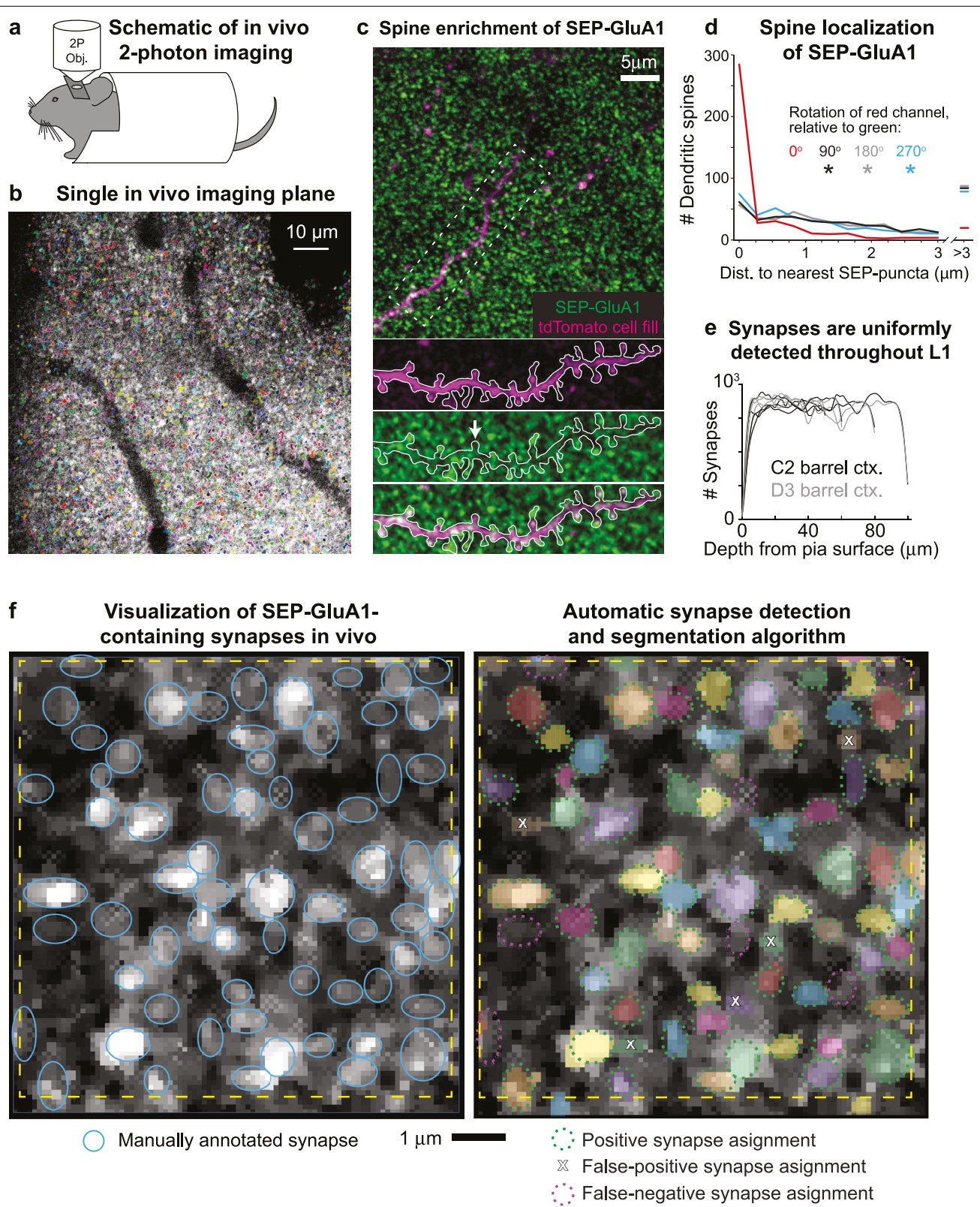

**Figure 7.** Visualizing SEP-GluA1 synapses in vivo using two-photon (2p) microscopy. (**a**) Schematic of in vivo 2p imaging. (**b**) Large-scale automatic detection and segmentation of SEP-GluA1-containing synapses in L1 barrel cortex. Automatically detected synapses are rendered in arbitrary colors. Dark areas likely correspond to either vasculature or cell bodies. (**c**) Single in vivo imaging plane showing SEP-GluA1 synapses (green) and a single layer 2/3 pyramidal cell filled with tdTomato (magenta). White arrow denotes a spine devoid of SEP-GluA1 signal. (**d**) SEP-GluA1 is enriched in dendritic

*Figure 7 continued on next page*

*Figure 7 continued*

spines. 78% of automatically detected dendritic spines (visualized using a sparse tdTomato cell fill) contained a SEP-GluA1 synapse, defined as edge-to-edge separation of red spines and green puncta <0.25 μm. This overlap occurred at a substantially higher rate than chance as the distance between spines (magenta channel) and their nearest SEP neighbor (green channel) significantly increased when the magenta channel was rotated either 90, 180, or 270° relative to green (n = 504 spines; *p<0.001; Mann–Whitney U test, relative to unrotated). (**e**) GluA1-containing synapses were uniformly detected throughout L1 barrel cortex, up to a depth of 100 μm below the pial surface. (**f**) Left: single in vivo imaging plane displaying raw, unprocessed SEP-GluA1 signal, taken 47 μm deep in layer 1 (L1) of barrel cortex. Putative GluA1-containing synapses are identified as bright puncta. Manual synaptic annotations are overlaid as blue ovals. Right: same cortical plane, but with automatically identified and segmented synapses rendered in arbitrary colors. Manual annotations are overlaid, recolored either green or magenta, corresponding to true positives (defined as >50% of total 3D voxels shared between manual and automatic annotations) or false negative (defined as manual annotations that did not overlap with an automatic detection), respectively; false positives (defined as automatically detected synapses that did not overlap with a manual annotation) are indicated by an X. Scale bar is 1 μm.

The online version of this article includes the following figure supplement(s) for figure 7:

**Figure supplement 1.** Rates of agreement and error for synapse detection methods.

**Figure supplement 2.** Schematic of in vitro automatic detection of Homer and SEP-GluA1 puncta.

**Figure supplement 3.** Radial and axial point spread function (PSF) of two-photon (2p) microscope.

Homer puncta were automatically detected in vitro using the same algorithm, with 50% of shared voxels defined as overlap. We found a true-positive rate of 75.6% (overlap of SEP-GluA1 and Homer), a 13.1% false-positive rate (SEP detected without Homer overlap), and an 11.3% false-negative rate (Homer detected without SEP overlap; *Figure 7—figure supplement 2e*). This false-negative rate is likely an overestimate of error as this fraction could correspond to Homer-containing synapses that either do not contain detectable levels of the GluA1 subunit (i.e., synapses where GluA2/3 heterodimers predominate) or are so-called silent synapses that do not contain any AMPARs at all.

To further validate our synapse algorithm, we examined overlap between automatically detected synapses and dendritic spines using a sparse cell fill. Enrichment of SEP-GluA1 in spine heads was readily apparent (*Figure 7c*) as 78% of dendritic spines (visualized using a sparse

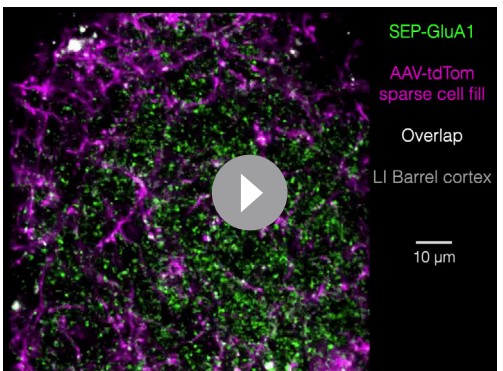

**Video 1.** Representative in vivo two-photon (2p) imaging volume of SEP-GluA1 knockin. The imaging volume displayed is 98 × 98 × 130 μm section of layer I barrel cortex, with SEP-GluA1 synapses in green and a sparse cell fill in magenta. Overlap between green/magenta voxels is rendered in white. The video begins at the pial surface, moving ventrally towards layer II/III, with depth indicated in the lower right. Note the high density of green puncta, each corresponding to a single GluA1-containing synapse. Punctate synaptic labels are observed uniformly up to a depth of ~100 μm, below which the signal begins to degrade, likely due to light scattering. In this SEP-GluA1 × Ai9 mouse, a random subset of layer II/III pyramidal cells was filled using a dilute (1:20k) AAV-CaMK2-Cre virus. Voxel size is 0.096 × 0.096 × 1 μm. Resolution is 1024 × 1024 pixels in XY and a 1 μm step size. Images were median filtered with a radius of 1 and contrast enhanced. Scale bar is 10 μm.

https://elifesciences.org/articles/66809/figures#video1

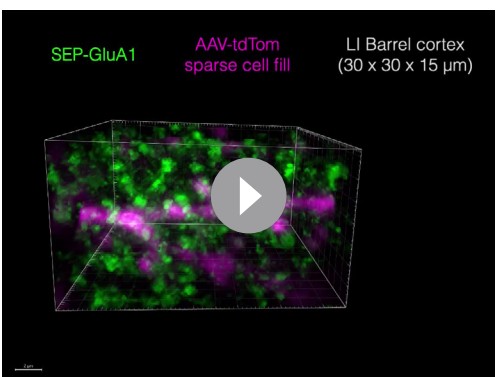

**Video 2.** 3D rendering of filled dendrite with SEP-GluA1 puncta in spines. The imaging volume displayed is a 30 × 30 × 15 μm section of layer I barrel cortex. Note the enrichment of SEP-GluA1 puncta (green) in dendritic spines (magenta) and the paucity of SEP signal in the dendritic shaft. There are many green puncta outside of the filled cell, likely corresponding to GluA1-containing synapses in spines of unlabeled cells. 3D volume was rendered using Imaris. Dynamic scale bar in lower left.

https://elifesciences.org/articles/66809/figures#video2

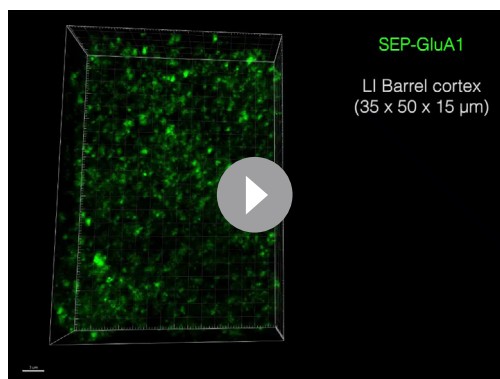

**Video 3.** 3D rendering of SEP-GluA1 puncta in layer I barrel cortex. 35 × 50 × 15 µm imaging volume. Note the extreme density of green puncta, each corresponding to a single GluA1-containing synapse. Rendered using Imaris. Dynamic scale bar in lower left. https://elifesciences.org/articles/66809/figures#video3

tdTomato cell fill) contained a SEP-GluA1 synapse, defined as edge-to-edge separation of red spines and green puncta <0.25 µm (*Figure 7c and d*). This overlap occurred at a substantially higher rate than chance as the distance between spines (magenta channel) and their nearest SEP-GluA1 neighbor (green channel) significantly increased when the magenta channel was rotated either 90, 180, or 270° relative to green (*Figure 7d*). SEP-GluA1 synapses were detected at a uniform rate across depth within layer 1 (L1) of barrel cortex, up to 100 µm below the pial surface (*Figure 7e*).

Our automatic synapse detection algorithm enables robust identification and segmentation of hundreds of thousands of SEP-GluA1 synapses in vivo, with accuracy at least comparable to expert human annotators, but with vastly increase speed and scale. Indeed, we observed similar accuracy and error rates between human annotated and automatically detected synapses in the same cortical volumes. Using the same threshold for defining overlap of 50% shared voxels (see *Figure 7—figure supplement 1*), we found 80.8% agreement between automatic and human annotated synapses (green dashed ovals in *Figure 7f*), with an 8.2% false-positive rate (automatic detection without an overlapping human annotation; white X in *Figure 7f*) and an 11.0% rate of false negative (human annotation without overlapping automatic detection, magenta dashed ovals in *Figure 7f* and *Figure 7—figure supplement 1*). These accuracy rates are slightly better than the rate of agreement between two expert humans annotating the same volume (72.3% agreement).

To accurately estimate our synaptic detection resolution boundaries, we measured the point spread function (PSF) of our 2p microscope (*Figure 7—figure supplement 3*). The PSF was reconstructed from averaged 2p images of 1 µm TetraSpeck microspheres, which is computed through an inverse deconvolution of raw images following established methods (*Padman et al., 2014*; *Verveer et al., 2020*). This process is necessary to extract the PSF from finite (nonzero) sized beads as using the raw bead images directly will give an overestimation of the PSF. The resolution of our 2p microscope in the X and Y directions is 0.55 and 0.57 µm, respectively (full width at half maximum [FWHM] of reconstructed PSF), compared to 2.50 µm in Z, suggesting that our ability to accurately segment boundaries of individual synapses in XY is greater compared to segmentation in Z. Given this relatively lower axial resolution inherent to 2p microscopy, we are unable to accurately segment two synapses that overlap in XY and are within three adjacent z-sections (each separated by 1 µm). Accordingly, we tuned our automated synapse detection algorithm to exclude synapses that we were unable to accurately segment because they closely abut in z (see rule 8, above) by discarding automatic detections that spanned more than six adjacent z-sections. This rule was implemented to minimize erroneous detection and segmentation of extremely large and high-total-intensity synapses arising from merge errors of two synapses closely opposed in z. Given the high density of synapses in cortex and the comparatively poor axial PSF of 2p microscopy, it is still possible that our automatically detected synapses do contain erroneous merge errors. However, a significant number of synapse merge errors would likely be observed as additional peaks in the histograms of synaptic intensity, corresponding to automatic detections containing, for example, two synapses occurring at 2× the max intensity peak of detections corresponding to a single synapse. We did not observe such peaks in the distribution of detected synapses (*Figure 8b*), supporting that our results accurately correspond to individual synapses, rather than two or more closely abutting synapses that were erroneously merged into one. Nevertheless, future studies could benefit from either post hoc immunocytochemical labeling of synaptic markers (e.g., Homer

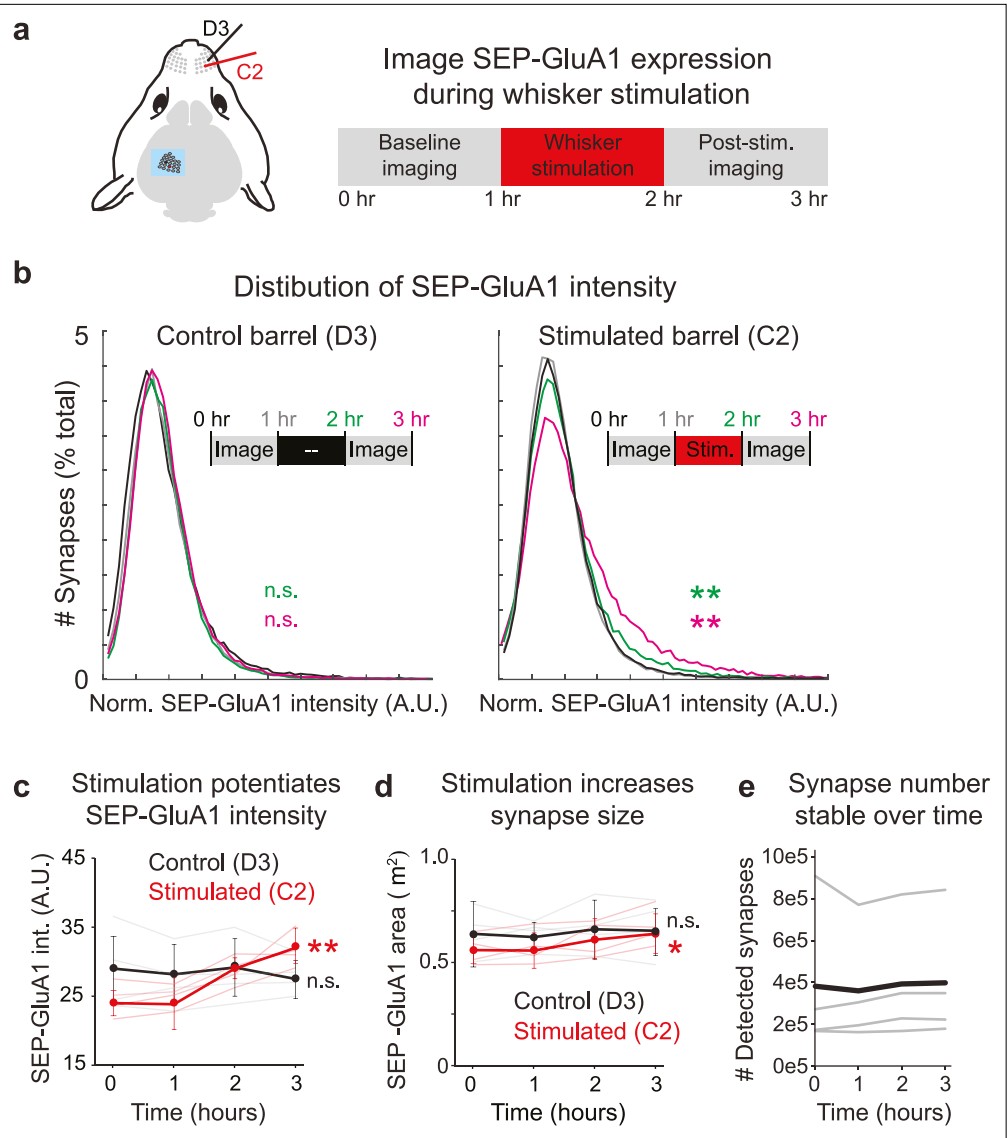

**Figure 8.** Tracking synaptic plasticity during sensory experience. (**a**) Schematic of whisker stimulation and in vivo imaging of barrel cortex. The control unstimulated (D3) and stimulated barrel (C2) were imaged twice at baseline. The C2 whisker was mechanically stimulated with 10 Hz vibration for 1 hr, after which imaging of both barrels resumed. (**b**) Distribution of normalized SEP-GluA1 intensity in barrel cortex over time in one representative homozygous SEP-GluA1 mouse. Left: distribution of SEP-GluA1 intensity was stable over time in the control, unstimulated D3 barrel. Right: significant rightward shift in SEP-GluA1 intensity in the C2 barrel following whisker stimulation (for 1 hr, between the 1 and 2 hr time points), indicating net synaptic potentiation. **$p<0.01$, Mann–Whitney U test relative to within-mouse baseline period (pooled 0 and 1 hr time points). (**c**) Whisker stimulation induces barrel-selective synaptic potentiation. Plot of mean SEP-GluA1 fluorescent intensity from all automatically detected SEP puncta over time in the control (black, D3) and stimulated (red, C2) barrel. n = 4 mice, **$p<0.01$, two-way ANOVA, comparing each imaging session to the two pooled baseline sessions (0 and 1 hr). Error bars represent standard deviation. (**d**) Whisker stimulation induces barrel-selective synapse enlargement. Plot of mean area of all automatically detected SEP puncta over time in the control (black, D3) and stimulated (red, C2) barrel. Synapse area was defined as the maximum area in a single 2D imaging plane for each automatically segmented SEP-GluA1 puncta. *$p<0.05$, two-way ANOVA, comparing each imaging session to the two pooled baseline sessions (0 and 1 hr). (**e**) Number of detected synapses was stable over time. Total synapse count from each individual mouse (thin gray) and mean (n = 4, thick black) are presented.

The online version of this article includes the following source data for figure 8:

**Source data 1.** Raw data used to make *Figure 8* plots.

or PSD95) or parallel in vivo labeling strategies to observe synaptic markers, to aid in defining the ground truth for synapse detection.

## SEP-GluA1 reports synaptic plasticity underlying sensory stimulation

To demonstrate the utility of the SEP-GluA1 knockin line and our automated synapse detection algorithm, we investigated synaptic dynamics in barrel cortex during whisker stimulation. Mouse somatosensory cortex displays an exquisite somatotopic map, wherein each individual whisker is represented by a discrete cortical area. These so-called barrels provide an ideal tableau to investigate activity-dependent plasticity underlying encoding of sensory stimulation. Previous work has shown that whisker stimulation can induce NMDA receptor-dependent LTP in layer 2/3 cells (*Gambino et al., 2014*; *Holtmaat and Caroni, 2016*; *Zhang et al., 2015*). To investigate how this sensory stimulation may be stored within vast synaptic networks, we surgically implanted cranial windows over barrel cortex in adult SEP-GluA1 mice (2–4 months old). Using optical-intrinsic imaging during passive whisker stimulation, we identified barrels corresponding to the C2 and D3 whiskers (*Figure 8a*). We imaged SEP-GluA1 fluorescence at high resolution within 100 µm cubed volumes of layer I somatosensory cortex. By registering volumes to vasculature and other fiducial markers, we were able to longitudinally image the same cortical volumes in each barrel for 3–5 hr. To confine our analysis to precisely the same neural volumes throughout sensory stimulation, we further employed post hoc rigid-body transformations to longitudinally align neural volumes.

We were able to detect hundreds of thousands of synapses in each mouse and extracted their SEP-GluA1 fluorescence intensity at each time point. To investigate synaptic dynamics at baseline and during sensory stimulation, we delivered 10 Hz mechanical stimulation exclusively to the C2 whisker in lightly anesthetized animals. Whereas the unstimulated barrel displayed a stable distribution of SEP-GluA1 intensity over time, the C2 barrel that received mechanical whisker stimulation displayed a significant rightward shift in the distribution of SEP-GluA1 intensity, consistent with increased synaptic SEP-GluA1 levels from induction of LTP (*Figure 8b–d*). This potentiation manifested as an increase in the mean SEP-GluA1 intensity (*Figure 8c*) as well as a smaller but still significant increase in synapse size (*Figure 8d*). The number of detected synapses in both the stimulated and unstimulated barrels was stable over time (*Figure 8e*), suggesting that the observed plasticity was not expressed via net spinogenesis or pruning, but rather by a net potentiation of existing synapses. As this knockin line exclusively labels GluA1-containing AMPARs, it is possible the changes we observed reflect subunit-specific dynamics rather than more general synaptic plasticity.

## Discussion

A central goal of neuroscience is to understand high-order cognitive functions in terms of their constituent components. Over the past 50 years, we have learned a great deal regarding the general role of synaptic plasticity in learning and memory. While seminal experiments have clearly implicated regulation of AMPARs as a central mechanism to modify the strength of synaptic communication between neurons (*Andersen et al., 1977*; *Frey and Morris, 1997*; *Huganir and Nicoll, 2013*; *Malinow and Malenka, 2002*; *McNaughton et al., 1978*), there is currently a dearth of methodologies to investigate how these molecular dynamics are distributed within vast networks of billions of synapses throughout the brain. For instance, patch-clamp recordings provide excellent spatial and temporal resolution, enabling investigation of how integration of specific synaptic inputs is dynamically tuned by plasticity (*Spruston, 2008*), but the scale of these recordings is limited to single neurons and performing them in behaving animals is challenging (*Bittner et al., 2017*; *Epsztein et al., 2011*). In vivo calcium imaging and high-channel-count electrophysiology offer superb spatial coverage, enabling investigation of neuronal activity within circuits of hundreds of neurons during behavior (*Juavinett et al., 2019*; *Jun et al., 2017*; *Sofroniew et al., 2016*), but these techniques lack the spatial resolution to study plasticity of individual synapses. In vivo structural imaging has provided valuable insights regarding how spine formation and elimination has been shown to contribute to neuronal development and synaptic plasticity (*Bhatt et al., 2009*; *Holtmaat and Svoboda, 2009*; *Trachtenberg et al., 2002*; *Xu et al., 2009*; *Yang et al., 2009*), though this method does not measure plasticity at existing spines. In addition, spine size has been reported to be proportional to synaptic strength (*Matsuzaki et al., 2001*), but this is at best an

indirect readout of synaptic strength, and in certain conditions spine size and synapse strength are completely dissociated (*Lee et al., 2012*; *Sdrulla and Linden, 2007*; *Tan et al., 2020*; *Zhang et al., 2015*). Thus, our understanding of how the brain represents learning, memory, and behavior is constrained by currently available methodologies.

Here, we present a suite of novel tools and approaches that break through these constraints, enabling visualization of synaptic plasticity with molecular resolution at brain-wide scale in living animals. At the heart of this suite lies the newly generated SEP-GluA1 knockin mouse, a novel line that fluorescently labels all endogenous GluA1-containing AMPARs throughout the entire brain. This line enables direct investigation of the molecular dynamics underlying synaptic plasticity at any scale, from super-resolution synaptic imaging in primary cultures to circuit-level analyses of plasticity in acute slices to brain region-wide imaging of synaptic strength in behaving animals. The sequence linking the SEP tag and AMPAR N-terminus is known to affect protein expression and proper postsynaptic targeting of the receptor as previous attempts to fluorescently tag AMPARs at the N-terminus have been reported to result in impaired synaptic function (*Díaz-Alonso and Nicoll, 2021*; *Díaz-Alonso et al., 2017*). However, in our experiments, N-terminal linkers were lengthened and optimized to increase the flexibility of the SEP tag, thereby limiting disruption of GluA1 function.

We conducted extensive validation of our novel knockin line, demonstrating that SEP-GluA1 mice exhibit normal synaptic physiology, AMPA receptor trafficking, and general behavior that is indistinguishable from WT littermates, strongly supporting that our endogenous labeling strategy does not impair synaptic function in any detectable manner. Using both primary cultures and acute slices, we showed that the SEP-GluA1 line reports both increases and decreases in synaptic GluA1 content, and it is an effective tool to study several forms of AMPAR-mediated plasticity, including homeostatic scaling and LTP. Using 2p glutamate uncaging, we clearly demonstrated that the intensity of SEP fluorescence directly correlates with functional synaptic strength, indicating that this line can be used as an effective tool to study synaptic plasticity in vivo. Using FRAP, we demonstrated that SEP-GluA1-containing synapses display normal synaptic mobility and dynamics. Finally, we developed a computer vision algorithm to automatically detect and segment extremely large numbers of endogenously labeled synapses across entire brain regions in living animals. Using these tools, we were able to longitudinally track synaptic plasticity encoding sensory stimulation with unprecedented spatial coverage and molecular resolution, producing the most detailed spatiotemporal map of behaviorally relevant synaptic plasticity to date. While the SEP-GluA1 line serves as an effective tool to investigate many forms of plasticity in vitro and in vivo, we are continuing to develop similar knockin and transgenic lines that similarly label GluA2-4, as well as other proteins of interest that may play key roles in synaptic transmission and plasticity. For example, it is important to consider that our approach to monitor SEP-GluA1 might favor the detection of changes in synapses preferentially undergoing plasticity rather than an absolute change in synaptic strength. Similar endogenous labeling strategies of other synaptic proteins would be useful to investigate the molecular mechanisms of nearly any behavior, including GluA1-independent forms of plasticity (*Frey et al., 2009*) or disease models that display synaptic pathologies, such as SynGAP haploinsufficiency or Alzheimer's disease (*Gamache et al., 2020*; *Sheng et al., 2012*).

By fluorescently tagging endogenous GluA1-containing AMPARs and utilizing in vivo 2p microscopy, we were able to directly visualize the functional strength of endogenous synaptic networks and track how they change during sensory stimulation. We demonstrated that mechanical stimulation of a single mouse whisker leads to increased synaptic GluA1 specifically in the cortical region corresponding to the stimulated whisker. This is consistent with previous studies showing NMDA receptor-dependent LTP following whisker stimulation (*Gambino et al., 2014*; *Holtmaat and Caroni, 2016*; *Zhang et al., 2015*). While our proof-of-principle experiments clearly illustrate the power of this approach to image endogenous AMPARs and track widespread synaptic plasticity in vivo, these data represent only the tip of the iceberg. As our genetic labeling strategy illuminates all GluA1-containing synapses throughout the brain, this line enables investigation of synaptic plasticity from any brain region during any behavioral paradigm of interest. Further, this flexible tool is compatible with any electrophysiological method or other fluorescence-based imaging approaches, such as neuronal activity sensors (e.g., RCaMP) and genetic tagging of cell types of interest (e.g., engram cells, Cre-lines, etc.). For example, by crossing the SEP-GluA1 knockin line with the Ai9 tdTomato

reporter mouse and expressing Cre in a cell-type or neuronal circuit of interest, it is possible to specifically quantify changes in spine GluA1 expression of these neurons, as demonstrated in *Figure 6* and *Figure 7c and d*. Beyond visualizing fluorescently labeled cortical synapses through cranial windows, one could also use our tools to investigate synaptic dynamics in subcortical structures using cortical excavation, endoscopes, or fiber photometry.

More generally, this strategy to label endogenous synaptic receptors has several key advantages over previous approaches. Building upon insights gleaned by spine dynamics, in which the formation and elimination of dendritic spines has been shown to be involved in several forms of learning (*Holtmaat and Svoboda, 2009*; *Xu et al., 2009*; *Yang et al., 2009*), our approach additionally enables investigation of plasticity in existing synapses. Recently, similar genetic labeling strategies have been used to investigate other synaptic proteins, such as PSD-95 (*Cane et al., 2014*; *Fortin et al., 2014*; *Gray et al., 2006*; *Zhu et al., 2018*). These studies have revealed crucial details regarding how scaffolding and structural proteins contribute to dynamic synapse function. Here, we build upon these findings by directly imaging AMPARs, which are the principal functional unit of the synapse. While overexpression of SEP-labeled AMPARs has been previously used to investigate behaviorally relevant plasticity (*Diering et al., 2017*; *El-Boustani et al., 2018*; *Miyamoto et al., 2021*; *Roth et al., 2020*; *Suresh and Dunaevsky, 2017*; *Tan et al., 2020*; *Zhang et al., 2015*), the current study is the first to engage labeling of endogenous receptors, which much more faithfully reports the direct physiological mechanisms of plasticity.

Recently, deep learning-based systems have achieved state-of-the-art performance in analyzing microscopy images (*Moen et al., 2019*). Many different architectures have been used, including our work detecting tau tangles with sliding windows to annotate single pixels (*Tward et al., 2020*), UNET (*Olaf Ronneberger and Brox, 2015*) for annotating larger blocks that has been implemented in Fiji (*Falk et al., 2019*), and other elaborations such as VNET (*Milletari et al., 2016*). Typically, trained networks assign class probabilities to each pixel, which are collected into larger objects based on connected components or watershed approaches available in standard packages, such as Fiji (*Schindelin et al., 2012*). A caveat of the deep learning approach is that these methods require a large quantity of high-quality training data, which is not available for new image types such as those used in our work. Another approach to annotating microscopy images has been to employ interactive methods. For example, the framework developed by ilastic allows users to annotate small regions and constructs a random forest model based on simple features (brightness, edges, texture) to extend the per pixel annotations to large images (*Berg et al., 2019*). The annotated regions can be modified or extended until optimal performance is achieved. Importantly, this approach works well in situations where annotation is easy 'by eye,' but is challenging at scale. Ultimately, we found that our rules-based, template-matching approach performed well at automatically detecting and segmenting synapses imaged in vivo, with true-positive, false-positive, and false-negative rates on par with or better than comparisons between two expert human annotators.

To achieve the goal of understanding plasticity at individual synapses during complex behaviors and learning, it is crucial to track individual synapses over time. Here, we have tracked the same population of spines during whisker stimulation (*Figure 8*) and have also shown that by adding a cytosolic fluorescent protein it is possible to achieve longitudinal imaging with the ability to track individual synapses across days (*Figure 6a and c*). In future studies, we will expand on our computational approach to enable alignment, registration, and tracking of millions of individual synapses throughout the entire process of learning. While our current imaging experiments were performed in lightly anesthetized mice, our previous work has demonstrated the feasibility to visualize and track SEP-GluA1 in individual spines in head-fixed awake behaving mice (*Tan et al., 2020*) with the same reliability and resolution as in anesthetized mice, suggesting that imaging awake SEP-GluA1 knockin mice will be feasible.

In conclusion, we aim to fundamentally advance our understanding of the synaptic basis of behavior, moving beyond merely studying synaptic plasticity in single neurons, seeking instead to explore dynamic modulation of the complete synaptome during learning and memory. The tools presented here make this goal achievable.

# Materials and methods

**Key resources table**

| Reagent type (species) or resource | Designation | Source or reference | Identifiers | Additional information |
|---|---|---|---|---|
| Antibody | Anti-GluA1 C-terminal (rabbit polyclonal) | *Oku and Huganir, 2013* | JH4294 | (1: 2000) Made in house Shared upon request |
| Antibody | Anti-GFP (chicken polyclonal) | Abcam | Ab13970 RRID:AB_300798 | (1:2000) |
| Antibody | Anti-Psd-95 (mouse monoclonal) | NeuroMab | Clone K28/43 Cat # 75-028 RRID:AB_2292909 | IF (1:500) WB (1:5000) |
| Antibody | Anti-Homer 1 (rabbit polyclonal) | Synaptic Systems | 160-003 | (1:1000) |
| Antibody | Anti-GluA2 N-terminal (mouse monoclonal) | This paper | 032.19.9 | (1:2000) Made in house Shared upon request |
| Antibody | Anti-GluA3 N-terminal (rabbit polyclonal) | This paper | JH4300 | (1:1000) Made in house Shared upon request |
| Cell line (*Mus musculus*) | SEP-GluA1 primary cultured neurons | This paper | SEP-GluA1 | Made in house Will deposit to Jackson Labs |
| Genetic reagent (*M. musculus*) | SEP-GluA1 knockin mice | This paper | SEP-GluA1 | Made in house Will deposit to Jackson Labs |
| Genetic reagent (*M. musculus*) | Ai9 (B6.Cg-Gt(ROSA)26Sortm9(CAG-tdTomato)Hze/J) mice | The Jackson Laboratory | Jax # 007909 RRID:IMSR_JAX:007909 | |
| Other | AAV-CaMKII-Cre virus | Addgene/Penn Vector | #105558-AAV1 | (1:10,000–1:50,000) |
| Software, algorithm | Fiji image processing software | Fiji | | |
| Software, algorithm | GraphPad Prism 9 | GraphPad Prism 9 | | |
| Software, algorithm | ANYmaze animal tracking software | Stoelting | | |
| Software, algorithm | Clampex 10.7 | Molecular Devices | | |
| Software, algorithm | Clampfit 10.7 | Molecular Devices | | |
| Software, algorithm | Mini Analysis Program v 6 | Synaptosoft Inc | | |
| Software, algorithm | Igor Pro 6.3 | WaveMetrics | | |
| Software, algorithm | In vivo synapse detector | This paper | | https://github.com/twardlab/synapse_labeling (copy archived at swh:1:rev:5a274f9cb8afbad23bea698f02e43418e136ca8d), *Graves et al., 2020* |
| Software, algorithm | Imaris 9.6.0 | Oxford Instruments | | |
| Software, algorithm | Matlab2020b | MathWorks | | |

*Continued on next page*

*Continued*

| Reagent type (species) or resource | Designation | Source or reference | Identifiers | Additional information |
|---|---|---|---|---|
| Software, algorithm | ScanImage | Vidrio Technologies | | |
| Software, algorithm | StackGPS | This paper | | https://github.com/ingiehong/StackGPS, (copy archived at swh:1:rev:60b7378461a650d86c20e4a4f7cfb2c5aff3f20a), *Hong, 2021* |
| Software, algorithm | ImageJ | ImageJ | | |

## Neuronal culture

Mouse embryonic (E18) cortical/hippocampal neurons were plated on poly-L-lysine-coated tissue culture dishes/glass coverslips at a density of 65,000 cells/cm$^2$/37,500 cells/cm$^2$ in NM5 medium (neurobasal media [Invitrogen] supplemented with 2% B-27, 2 mM GlutaMAX, 50 U/mL PenStrep, and 5% horse serum [Invitrogen]) and grown in NM0 medium (neurobasal media [Invitrogen] supplemented with 2% B-27, 2 mM GlutaMAX [50 U/mL, PenStrep]). Cultured cortical neurons/hippocampal neurons were fed twice/once per week. To induce synaptic scaling, cortical neurons were treated with bicuculline (20 µM) or TTX (1 µM) at DIV 11–13 for 48 hr. Hippocampal neurons were used at DIV 19–22 for glutamate uncaging.

## Surface biotinylation

Neurons were rinsed with ice-cold PBSCM (1× PBS, 1 mM MgCl$_2$, 0.1 mM CaCl$_2$, pH 8.0) once and then incubated with Sulfo-NHS-SS-biotin (0.5 mg/mL, Thermo Scientific) for 30 min at 4°C. Residual unreacted biotinylation reagent was washed out with PBSCM and quenched by 20 mM glycine twice for 5 min. Neurons were lysed in lysis buffer (PBS containing 50 mM NaF, 5 mM sodium pyrophosphate, 1% NP-40, 1% sodium deoxycholate, 0.02% SDS, and protease inhibitor cocktail [Roche]). 20 µg lysates were incubated overnight with NeutraAvidin agarose beads (Thermo Scientific) and then were washed with lysis buffer four times. Biotinylated proteins were eluted using 2× SDS loading buffer. Surface proteins were then subjected to SDS-PAGE and analyzed by western blot.

## PSD fractionation

Mouse hippocampus tissues were homogenized in buffer (320 mM sucrose, 5 mM sodium pyrophosphate, 1 mM EDTA, 10 mM HEPES pH 7.4, 200 nM okadaic acid, protease inhibitor cocktail [Roche]) using a 26-gauge needle. Homogenate was centrifuged at 800× g for 10 min at 4°C to yield P1 (nuclear) and S1 (post-nuclear). S1 was centrifuged at 20,000× g for 20 min to yield P2 (membrane) and S2 (cytosol). P2 was then resuspended in water adjusted to 4 mM HEPES pH 7.4 followed by 30 min agitation at 4°C. Suspended P2 was centrifuged at 25,000× g for 20 min at 4°C. The resulted pellet was resuspended in 50 mM HEPES pH 7.4, mixed with an equal volume of 1% Triton X-100, and agitated at 4°C for 10 min. The PSD fraction was generated by centrifugation at 32,000 × g for 20 min at 4°C.

## Cell-culture immunohistochemistry and confocal imaging

Cultured hippocampal neurons were fixed for 20 min in PBS containing 4% paraformaldehyde (PFA)/4% sucrose and rinsed with PBS. Neurons were blocked, permeabilized, and incubated with primary antibodies in GDB buffer (15 mM phosphate buffer [pH 7.4] containing 0.1% gelatin, 0.3% Triton X-100, and 0.25 M NaCl) at 4°C overnight. Coverslips were washed with PBS before the neurons were incubated with secondary antibodies in GDB buffer for 1 hr at room temperature. After washing with PBS and water, coverslips were mounted onto glass slides using Permafluor (Fisher Scientific). Images were obtained using an LSM880 laser scanning confocal microscope (Zeiss). The following antibodies were used: anti-GluA1 C-terminal pAb (JH4294, made in-house), anti-PSD95 mAb (NeuroMab), anti-GFP pAb (ab13970, Abcam), Alexa Fluor 488 goat anti-chicken (Thermo Fisher Scientific), Alexa Fluor 405 goat anti-mouse (Thermo Fisher Scientific), and Alexa Fluor 647 goat anti-rabbit (Thermo Fisher).

## Electrophysiological recordings

Whole-cell voltage-clamp recordings were performed in CA1 pyramidal neurons of acute hippocampal slices from 3- to 4-week-old paired littermates of mice by an experimenter blind to genotype. Slices

were prepared in ice-cold oxygenated dissection buffer containing the following (in mM): 210 sucrose, 7 glucose, 26.2 NaHCO$_3$, 2.5 KCl, 1 NaH$_2$PO$_4$, 7MgSO$_4$. For all recordings, slices were perfused in ACSF (119 mM NaCl, 26.2 mM NaHCO$_3$ and 11 mM glucose, 2.5 mM KCl, 1 mM NaH$_2$PO$_4$, 2.5 mM CaCl$_2$, 1.3 mM MgSO$_4$, 50–100 µM Picrotoxin) at room temperature. Neurons were patched by glass pipettes (3–5 MΩ), which were filled with internal solution (115 mM Cs-MeSO$_3$, 0.4 mM EGTA, 5 mM TEA-Cl, 2.8 mM NaCl, 20 mM HEPES, 3 mM Mg-ATP, 0.5 mM Na2-GTP, 10 mM Na phosphocreatine, 5 mM QX-314). mEPSC recordings were performed in the ACSF in presence of 1 µM TTX, and cells were held at –70 mV. Data from 5 to 10 min after break-in were used for mEPSC analysis. For rectification and LTP experiments, EPSCs were elicited at 0.1 Hz by electrical stimulation (0.1 ms, 8–20 µA) via a stimulating electrode positioned in stratum radiatum. During rectification measurements, 100 nM spermine was added into the internal solution and cells were held at –60–60 mV before liquid junction modification. Each data point at each potential was averaged by 5–10 EPSCs. LTP was induced by a train of 200 pulses at 2 Hz paired with 0 mV depolarization. Data are presented as EPSC amplitude averaged at 1 min intervals and normalized to baseline. Signals were measured with MultiClamp 700B amplifier and digitized at 10 kHz by using a Digidata 1440A. Data acquisition was performed with pClamp 10.5 software. Access resistance (Ra) was monitored throughout the recording. Cells in which the Ra >20 MΩ or Ra varied by more than 20% were discarded.

## Mouse behavior

Behavioral testing was performed in homozygous SEP-GluA1 mice ( nine females and seven males) and WT ( nine females and nine males) littermate controls, aged 6–10 weeks. Animals were housed in a holding room on a reverse light cycle, and testing was conducted during the dark (i.e., active) phase. All behavioral experiments were approved by the Johns Hopkins Johns Hopkins Animal Care and Use Committee.

Locomotor activity was assessed by placing animals in an illuminated open arena (40 × 40 cm) and measuring the number of infrared beam breaks during a 30 min session (San Diego Instruments Inc). Anxiety was assessed using an elevated plus maze (66 cm long and 5 cm wide; San Diego Instruments Inc), consisting of two closed arms and two open arms suspended 54 cm above the ground. Immediately before testing, animals were placed, individually, into a clean cage for 5 min. Animals were placed onto the center of the elevated plus maze facing an open arm and allowed to explore for 5 min. Animal position was tracked using ANYmaze software (Stoelting, IL).

Spatial short-term memory was assessed by testing spatial novelty preference using a Y-maze. The Y-maze was made of clear plexiglass (each arm 38 cm long; San Diego Instruments Inc) and surrounded by distal spatial cues. A mixture of clean and dirty sawdust (ratio 2:1) was added to the bottom of the maze to promote exploration of the maze. The dirty sawdust was collected from other cages mice of the same sex as the animals being tested. Immediately before testing, animals were placed, individually, into a clean cage for 5 min. The test was split into exposure and test phases. During the exposure phase, one of the Y-maze arms was blocked (counterbalanced for genotype) and animals were allowed to explore two arms of the maze for 5 min. After this exposure phase, animals were gently removed from the maze and returned to the temporary holding cage for 1 min during which the sawdust was redistributed and all arms of the maze were made available. For the test phase, mice were re-exposed to the maze and allowed to explore all arms for 2 min. Testing was conducted by an experimenter blind to genotype of the mice being tested. Statistical comparisons were made using SPSS (IBM). Sex and genotype were used as between-subject variables.

## 2p glutamate-uncaging

Cultured mouse cortical neurons (10:1 mixture of WT and homozygous SEP-GluA1) were plated at E18 and imaged on DIV 16–18. Neurons were perfused in a modified HEPES-based ACSF solution, consisting of (in mM): 140 NaCl, 5 KCl, 10 glucose, 10 HEPES, 2 CaCl$_2$, 1 MgCl$_2$, 1 TTX, and 2.5 mM MNI-caged-L-glutamate (Tocris), pH = 7.30 and 310–316 mOsm. Recordings were made at room temperature in recirculated ACSF (3 mL/min). Recording pipettes were fabricated (Flaming/Brown Micropipette Puller, Sutter Instruments) from borosilicate capillary glass (Sutter, 4–6 MΩ open-tip resistance) and filled with (in mM): 115 CsMeSO$_4$, 2.8 NaCl, 5 TEACl, 0.4 EGTA, 20 HEPES, 3 MgATP, 0.5 NaGTP, 10 Na phosphocreatine, and 2.5 QX-314, pH = 7.32 and 306 mOsm, and containing a 1%

Alexa-594 dye (Tocris). Whole-cell voltage-clamp recordings were made using a MultiClamp 700B amplifier and Digidata 1440A digitizer (Axon Instruments).

Neurons were imaged with a 20×/1.0 NA water-immersion objective (Zeiss) and a custom-built 2p microscope (MOM system, Sutter Instruments) controlled by ScanImage (Vidrio Technologies, Ashburn, VA). Dendritic morphology was visualized using an Alexa dye, delivered by the patch pipette. SEP-GluA1 and red cell fill were excited at 910 nm using a tunable Ti:sapphire laser (Coherent, Santa Clara, CA). Images were acquired at 1024 × 1024 resolution and slices within z-stacks spaced every 0.5 µm. A second 2p laser (Spectra Physics, Santa Clara, CA) was used to uncage glutamate (1 ms pulse) onto visually identified spines at a wavelength of 730 nm and a power of 20 mW at the objective back aperture. Uncaging position was controlled using custom software developed in our lab (Scan-Stim), which provided means to correct for chromatic aberration between the imaging and uncaging beam. The offset between the imaging and uncaging 2p lasers was directly measured and corrected on a monthly basis. To measure the glutamate uEPSC, we used pClamp (Axon Instruments) to synchronize triggering of the uncaging laser with voltage-clamp recordings. To minimize the effect of electrotonic filtering caused by variable numbers of branch points between the site of dendritic uncaging and the somatic recording pipette, we uncaged exclusively onto spines of secondary dendrites, located 95–160 µm from the cell body. We uncaged on 4–8 spines/dendritic segment and 1–3 dendritic segments/neuron. To quantify the SEP-GluA1 and cell-fill signals, we manually drew regions of interest (ROIs) around visually identified spines, summed the fluorescent intensity of five adjacent z-sections (each separated by 0.5 µm), and subtracted size-matched neighboring background ROIs. Representative images shown in figures were median filtered and contrast enhanced. The uncaging-LTP-induction stimulus consisted of 30 pairings of glutamate uncaging (1 ms pulse of 730 nm laser at 0.5 Hz) and postsynaptic depolarization (0 mV for 0.5 s, beginning concurrently with uncaging pulse). Spines were imaged every 5 min, and synaptic strength was probed by measuring the uEPSC amplitude of each identified spine every 1 min.

## Cranial window surgery and viral injection

Mice were anesthetized (2% isoflurane) and implanted with a 3 × 3 mm cranial window (Potomac Photonics) over the barrel cortex region of somatosensory cortex at 2–3 months of age. Windows were sealed and custom-made metal head bars attached using dental cement (Metabond; Edgewood, NY). In a subset of experiments, an AAV-CaMKII-cre virus (Addgene/Penn Vector) was injected into barrel cortex (1:10k –1:50k dilution, 100–150 nL, 0.25–0.3 mm deep) of double homozygous SEP-GluA1 × Ai9 reporter mice to sparsely label L2/3 pyramidal neurons with a tdTomato cell fill. 10 mg/kg of extended-release buprenorphine (ZooPharm) was administered before surgery and mice were observed for 3 days following surgery. Mice were allowed to recover from surgery for at least 2 weeks before commencing in vivo imaging. All surgical procedures were approved by the Johns Hopkins Johns Hopkins Animal Care and Use Committee.

## Optical-intrinsic and in vivo 2p imaging

Optical-intrinsic imaging was used to map select barrels within somatosensory cortex as previously described (*Zhang et al., 2015*). Briefly, we mechanically stimulated the individual C2 and D3 whiskers using a custom-built piezo driver at 10 Hz and used optical-intrinsic imaging to identify the corresponding barrel fields. Mice were anesthetized and maintained on 0.5% isoflurane supplemented by xylazine (13 mg/kg). Optical images of barrel cortex were acquired at 30 Hz using a CCD camera (Grasshopper GS3-U3-23S6M-C under red LED light [630 nm] with a 2.5×/0.075 numerical aperture (NA) objective [Zeiss]). Images were collected, averaged (across 30 trials), Gaussian filtered (σ = 10 µm), and baseline subtracted. Widefield images of both barrels were acquired and vasculature was used to align subsequent 2p imaging of the same regions.

In vivo 2p images were acquired from lightly anesthetized mice (13 mg/kg xylazine and 0.5% isoflurane) using a custom-built, 2p laser scanning microscope controlled by ScanImage (Vidrio Technologies) and a 20×/1.0 NA water-immersion objective lens (Zeiss). SEP-GluA1 (green) and tdTomato cell fill (red) were both excited at 910 nm with a Ti:sapphire laser (Spectra-Physics, 20 mW power at objective back aperture). Green and red fluorescence signals were acquired simultaneously and separated by a set of dichroic mirrors and filters (ET525/50m for green channel, ET605/70m for red channel, Chroma). Image stacks were acquired at 1024 × 1024 pixels with a pixel size of 0.096 µm in

XY, with a z-step of 1 μm. Representative images shown in figures were median filtered (1-pixel radius) and contrast enhanced.

## Fluorescence recovery after photobleaching in vivo

To longitudinally image the same populations of SEP-labeled synapses throughout FRAP, imaging volumes (1024 × 1024 pixels in XY, 15 μm in z with 1 μm steps) were manually aligned before each time point using the sparse tdTomato cell fill as a guide. Photobleaching of spines was achieved with repetitive xy scanning of specific ROIs (~22 × 22 pixels) defined at the center plane of the image stack using ROI Group Editor in ScanImage. Bleached subregions were excited at 910 nm with high-intensity illumination (20–30%) with a Ti:sapphire laser (Coherent, 15–100 mW of power delivered to the objective back aperture) at a dwell time of ~3 μs/pixel and seven iterations. For each experiment, ~5–10 spines were bleached at a time.

Frame alignment and averaging within each plane in the Z-stacks were performed using a rigid registration custom script, and images at different time points were aligned using StackGPS *Hong, 2021*, (https://github.com/ingiehong/StackGPS) in MATLAB. Fluorescence intensity values were measured in ImageJ. Circular ROIs were defined around bleached and unbleached control spines, and signal intensity was measured as the average of three planes centered on each spine. Values were background subtracted. Baseline fluorescence was normalized to 1, and the signal intensity of the bleached spines was normalized to the averaged signal intensity of the unbleached spines on the same image. FRAP was calculated as the fluorescence increase between time 0, immediately after photobleaching, and the indicated time points. Only spines that maintained stable levels of tdTomato signal in all imaging sessions after recovering from photobleaching (intensity signal above maximum photobleaching induced in that spine at time 0) were included in the analysis. Graphing and curve fitting were performed in Prism 6 (GraphPad software). Symbols represent mean, and error bars represent standard error of the mean (SEM). Curve fitting of fluorescence recovery from 0 to 30 min was performed using nonlinear regression to fit an exponential one-phase decay curve defined by $Y = (Y0-Plateau) * exp(-k*x) + Plateau$, where Plateau is the maximum fluorescence, Y0 is starting fluorescence, k is the rate constant of recovery (minutes-1), and x is time (min). Outlier removal was performed using the ROUT method with false detection rate Q = 1%. Solid line in curve fitting represents best fit curve, and shaded area represents the 95% confidence interval of the best fit.

## Automatic synapse detection and segmentation

We defined synapses as regions with bright centers and dark surrounds, based on the following method. Images were blurred with a Gaussian kernel of standard deviation 5 × 5 × 1 pixels, and all local maxima were considered to be candidates for synapses. Candidates less than 3 pixels from each other were removed using a farthest first traversal (*Hochbaum and Shmoys, 1985*; *Rosenkrantz et al., 1977*). A family of templates (*Brunelli, 2009*) were defined using square regions with a radius of 32 pixels. The foreground was described by ellipses containing between 20 and 150 pixels, 4 roundnesses (ratio of larger to smaller semimajor axes lengths) from 1 to 2.5, and 12 angles from 0 to 2 pi. A background region of width 3 pixels was identified surrounding the ellipse. For each candidate synapse, an SNR was calculated to determine the most likely template: mean pixel intensity of foreground, minus mean of background, divided by the standard deviation of the region. The template that maximized SNR was associated to this candidate. Regions lacking detectable SEP fluorescence, such as blood vessels and cell bodies, were excluded by thresholding, removing areas with a z score of less than –1 in a blurred image. Based on visual examination, these regions were most likely interneuron cell bodies or blood vessels.

Candidates were accepted as synapses only if all five of the following conditions were met: (1) their associated template had a size between 20 and 150 pixels (in a single Z-plane); (2) ovacity between 1 and 2.5; (3) SNR greater than the 90th percentile of 300 randomly selected locations; (4) SNR was not reduced by more than 33% when averaging two adjacent slices; and (5) SNR was not increased by averaging seven adjacent slices. The fourth criterion was chosen because synapses span more than one slice, whereas noise does not, and the fifth was chosen because artifacts (e.g., autofluorescence) tend to span many more slices than synapses do. Candidate synapses in adjacent Z-planes that overlapped in XY were merged into a single, 3D synapse volume. To be finally considered a valid synapse,

two further criteria were required: (1) 3D volumes must contain XY-overlapping putative synapses on 2–6 adjacent planes in z.

Detection code was written in Python using numpy and is made available in the form of a Jupyter notebook at https://github.com/twardlab/synapse_labeling.

We defined agreement between two annotated sets of synapses, A and B, as follows. For each synapse in set A, we identified any overlapping synapses in B. If more than one overlapped, we chose one that overlapped by the largest amount. If there were no overlapping synapse, or the overlapping synapse in B overlapped by less than 50% the size of the synapse in A, this was considered a disagreement. Otherwise, voxel overlap of more than 50% was considered agreement. The accuracy between A and B was defined as the fraction of synapses that agreed. The accuracy between B and A was defined by reversing the roles of A and B above. This definition is not symmetric, and the two agreements are generally close but not equal. In each relevant figure, we report the average A-B and B-A agreement.

## Ground-truth synapse detection using immunohistochemistry

To label, visualize, and detect an independent synapse channel to which to compare SEP-GluA1, we perfused 10-week-old homozygous SEP-GluA1 mice with 4% PFA, made 100-μm-thick slices of barrel cortex, and stained for the PSD protein Homer using a polyclonal antibody (Synaptic Systems) and an Alexa-555 secondary antibody. Stained tissue was imaged with the same 2p microscope and light path as used for in vivo imaging, using identical settings for excitation wavelength (910 nm), laser power, PMT gain, scan speed, and all other acquisition settings. Green (SEP-GluA1) and red (Homer) channels were separately analyzed using our in vivo automatic synapse detection algorithm. Ground truth for synapse detection was defined as the rate of overlap between SEP-GluA1 and Homer puncta, false positive was defined as the rate of SEP-GluA1 detection without overlapping Homer, and false negative was defined as the rate of Homer detection without overlapping SEP-GluA1. A threshold of >50% shared voxels was used to assess overlap for all comparisons.

## Acknowledgements

ARG was supported by R21 AG063193. ARG and DJT were supported by Kavli Distinguished Post-doctoral Fellowships. ARG, AMB, and RLH were supported by R01 MH123212. AMB was supported by K99MH124920. ARG, JTV, and RLH were supported by a Schmidt Science Nascent Innovation Grant. ACS was supported by a Kavli Distinguished Graduate Fellowship. We thank members of the Huganir Lab for helpful comments and discussion.

## Additional information

### Competing interests

Michael I Miller: Dr. Miller is a joint owner of AnatomyWorks. Dr. Miller's relationship with Anatomy-Works is being handled under full disclosure by the Johns Hopkins University. The other authors declare that no competing interests exist.

### Funding

| Funder | Grant reference number | Author |
| --- | --- | --- |
| National Institutes of Health | R21 AG063193 | Austin R Graves<br>Richard L Huganir |
| Kavli Foundation | | Austin R Graves<br>Alina C Spiegel<br>Daniel J Tward |
| National Institutes of Health | R01 MH123212 | Austin R Graves<br>Alexei M Bygrave<br>Michael I Miller<br>Richard L Huganir |

| Funder | Grant reference number | Author |
|---|---|---|
| Schmidt Science Nascent Innovation Grant | 1 | Austin R Graves Joshua T Vogelstein Richard L Huganir |
| National Institutes of Health | K99 MH124920 | Alexei M Bygrave |

The funders had no role in study design, data collection and interpretation, or the decision to submit the work for publication.

### Author contributions
Austin R Graves, Conceptualization, Data curation, Formal analysis, Funding acquisition, Investigation, Validation, Visualization, Writing - original draft, Writing - review and editing; Richard H Roth, Han L Tan, Qianwen Zhu, Alexei M Bygrave, Elena Lopez-Ortega, Conceptualization, Data curation, Formal analysis, Investigation, Validation, Visualization, Writing - review and editing; Ingie Hong, Resources, Software, Writing - review and editing; Alina C Spiegel, Formal analysis, Investigation, Visualization, Writing - review and editing; Richard C Johnson, Methodology, Resources, Validation; Joshua T Vogelstein, Michael I Miller, Funding acquisition, Supervision; Daniel J Tward, Formal analysis, Methodology, Resources, Software, Visualization, Writing - review and editing; Richard L Huganir, Conceptualization, Funding acquisition, Project administration, Supervision, Writing - review and editing

### Author ORCIDs
Austin R Graves  http://orcid.org/0000-0003-1087-3684
Richard H Roth  http://orcid.org/0000-0002-6855-999X
Han L Tan  http://orcid.org/0000-0001-5163-7720
Alexei M Bygrave  http://orcid.org/0000-0003-2291-923X
Ingie Hong  http://orcid.org/0000-0002-7246-9233
Joshua T Vogelstein  http://orcid.org/0000-0003-2487-6237
Richard L Huganir  http://orcid.org/0000-0001-9783-5183

### Ethics
These studies were conducted in accordance with US Public Health Service on Human Care and Use of Laboratory Animals (PHS Policy) and all procedures involving animals were approved by the Johns Hopkins Animal Care and Use Committee (ACUC) protocols (MO19M274, MO20M372, MO20M92, MO20M336). All surgeries were performed under isoflurane anesthesia. Every effort was made to reduce or eliminate pain and suffering during all surgical procedures, in vivo imaging sessions, and behavioral experiments.

### Decision letter and Author response
Decision letter https://doi.org/10.7554/eLife.66809.sa1
Author response https://doi.org/10.7554/eLife.66809.sa2

## Additional files

### Supplementary files
• Transparent reporting form

### Data availability
We have provided Source data for our Figures and deposited the remaining data to Dryad with https://doi.org/10.5061/dryad.ttdz08m0b. All code used to analyze and process data is freely available on GitHub, with links specified in the manuscript.

The following dataset was generated:

| Author(s) | Year | Dataset title | Dataset URL | Database and Identifier |
|---|---|---|---|---|
| Huganir RL, Graves A | 2021 | Raw data for Graves et. al 2021 eLife | https://doi.org/10.5061/dryad.ttdz08m0b | Dryad Digital Repository, 10.5061/dryad.ttdz08m0b |

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
