## [Decision Letter]

Thank you for submitting your article "Visualizing synaptic plasticity in vivo by large-scale imaging of endogenous AMPA receptors" for consideration by *eLife*. Your article has been reviewed by 3 peer reviewers, including Brice Bathellier as the Reviewing Editor and Reviewer #1, and the evaluation has been overseen by Lu Chen as the Senior Editor.

Essential revisions:

1) The authors should clarify how the synapse detection algorithm account for extrasynaptic GluA1 labeling. Increases in this pool may not directly underlie increases at this synapse. Testing these confounding factors could be performed using physiologically relevant plasticity protocols in acute brain slices rather than cell culture, a preparation in which the neuron type and connectivity may not be the same as the mostly intact circuit.

2) The extended z-resolution of the 2P point-spread-function (PSF) in vivo will likely falsely assign fluorescence from multiple synapses to a single punctum. How does this limited resolution impact the final interpretations? The study would benefit from an in vivo ground-truth estimate of synapse location to confirm that SEP-GluA1 puncta correspond to an increase in AMPARs at synapses. The use of Homer does not entirely circumvent the uncertainty generated by the poor resolution of 2P. Post-hoc confocal microscopy in fixed tissue using morphology and synaptic markers could be sufficient. If not, then super-resolution fluorescence or immunogold EM are certainly suitable ground-truth approaches.

3) The quantification of synaptic strength by SEP-GluA1 may be an overestimate because SEP-GluA1 containing AMARs represents a fraction of the total receptor population, and LTP is strongly biased towards GluA1 insertion. How can this potential error be overcome?

4) To verify that SEP-GluA1 can reliably report decreases in synaptic strength, the authors should examine the activity-dependent endocytosis of receptors following LTD pairing protocols, preferably from the same cell type in the brain slices.

5) The choice to fluorescently label GluA1 subunits of the AMPAR to mark plasticity is, on the one hand, an obvious choice because of its implication in LTP. However, evidence in the literature suggests that LTP in the somatosensory cortex can be GluA1-independent (e.g., Frey et al., 2009). The authors should discuss these studies' implications and the limits of their method for addressing synaptic mechanisms for learning and memory. Does SEP-GluA1 fluorescence reproduce previous estimates of the distribution of synaptic weights? The SEP-GluA1 method is undoubtedly more attractive than the post hoc patch-clamp-based approaches performed by Ko et al. 2011, 2013 (Hofer and Mrsic-Flogel laboratories), but SEP-GluA1 should be compared to spine intensity (volume) imaging for example.

6) Conditional expression of the SEP-GluA1 would allow the precise dissection of synaptic plasticity specific neuron types within microcircuits, thereby linking plasticity with altered circuit function and behavior. This limitation must be addressed, in detail, in the Discussion. The current data sets of whole-brain expression of SEP-GluA1 do not distinguish between layer 2/3 pyramidal neurons, layer 5 pyramidal neurons, or interneurons. It is interesting to consider the possibility that a cross between the SEP-GluA1 transgenic mouse and a PSD-95 YPF-Enabled (Fortin et al., 2014) mouse could be used to assess the GluA1 changes at specific synapse types.

7) Using Western Blot, the authors show a 57% reduction of GluA1 expression at the postsynaptic density relative to WT, however, the potential implications of this are not addressed. In figure 1f, they show relative AMPA levels resulting from their Western blot analysis. Still, they do not adequately describe how the Western blot and analysis were performed, and namely, what control they used to normalize protein levels between different fractions. It would be nice to see additional statistical analyses of GluA2 and GluA3 expression relative to WT as there seems to be an increase and, therefore, potential compensation by GluA3 subunits SEP-GluA1 KI.

8) In the FRAP experiment in Figure 6, it seems like there could be other interpretations for the time course of recovery of fluorescence. The expected timescales of fluorescence recovery and caveats of its interpretations should be discussed.

9) An essential aspect of plasticity in the brain is that it can occur on long time scale on precise synapses. Here the authors do not show that it is possible to do longitudinal studies in which individual synapses are tracked. The authors need to provide some data to allow evaluating if this this is actually possible with their method. e.g. with a cytoplasmic marker for spines.

10) Comparison of their synapse detection algorithm with concurrent methods is necessary to establish it as a new tool.

*Reviewer #1 (Recommendations for the authors):*

1) Better discuss the limits of synapse identification on images. The limits are related both to the algorithm used and to 2-phoron imaging resolution limits. Thus better identification could be obtained maybe with super resolution methods. Ideally providing ground truth data would be an asset.

2) An essential aspect of plasticity in the brain is that it can occur on long time scale on precise synapses. Here the authors do not show that it is possible to do longitudinal studies in which individual synapses are tracked. I recommend to show some data that it is actually possible with their method, e.g. with a cytoplasmic marker for spines.

3) Comparison with previous methods (e.g. spine volume imaging) is necessary to establish this tool as a useful to track synaptic plasticity. This could be done in the whisker-based plasticity assay.

*Reviewer #2 (Recommendations for the authors):*

This is an important paper with well carried-out experiments. The conclusions are well supported by data. I have however 3 concerns (that do not dampen my enthusiasm):

1) The global expression of GluA2 is twice as low as the expression of GluA1 in wild-type mice, although the level of GluA2 remains unchanged. How this would affect GluA1/Glu2 heteromers? In addition, can the authors exclude the possibility that the decrease of SEP-GluA1 is brain region specific? (e.g. different levels of SEP-GluA1 between the hippocampus and the cortex). if the decrease is indeed region-specific, could the author exclude that physiological deficit does not occur in other brain region? At best, the authors should perform a better comparison of GluA1 and SEP-GluA1 between brain regions. A least, these point should be discussed. In the same line, bidirectional changes after Bic and TTX treatment in culture are really convincing (Figure 3B). However, it would be fair to present also the non-normalized level of surface AMPAR after treatment as compared to WT.

2) Although the authors present strong and convincing evidence that the high density of SEP-GluA1 synapses does not preclude the report of synaptic plasticity in anesthetized mice, it is much less clear whether it would work also in awake mice.

3) How effective is their supervised detecting method as compared to existing computer-based detection methods ? (e.g. see figure 7 from: https://www.biorxiv.org/content/10.1101/061507v2.full.pdf). Also, how robust is the supervised annotation of spines when the same FOV is imaged chronically (between days)?

*Reviewer #3 (Recommendations for the authors):*

My specific concerns are as follows:

1) How does the synapse detection algorithm account for extrasynaptic GluA1 labeling? Increases in this pool may not directly underlie increases at this synapse. Testing these confounding factors could be performed using physiologically relevant plasticity protocols in acute brain slices rather than cell culture, a preparation in which the neuron type and connectivity may not be the same as the mostly intact circuit.

2) The extended z-resolution of the 2P point-spread-function (PSF) in vivo will likely falsely assign fluorescence from multiple synapses to a single punctum. How does this limited resolution impact the final interpretations? The study would benefit from an in vivo ground-truth estimate of synapse location to confirm that SEP-GluA1 puncta correspond to an increase in AMPARs at synapses. The use of Homer does not entirely circumvent the uncertainty generated by the poor resolution of 2P. Post-hoc confocal microscopy in fixed tissue using morphology and synaptic markers could be sufficient. If not, then super-resolution fluorescence or immunogold EM are certainly suitable ground-truth approaches.

3) The quantification of synaptic strength by SEP-GluA1 may be an overestimate because SEP-GluA1 containing AMARs represents a fraction of the total receptor population, and LTP is strongly biased towards GluA1 insertion. How can this potential error be overcome?

4) To verify that SEP-GluA1 can reliably report decreases in synaptic strength, the authors should examine the activity-dependent endocytosis of receptors following LTD pairing protocols, preferably from the same cell type in the brain slices.

5) The choice to fluorescently label GluA1 subunits of the AMPAR to mark plasticity is, on the one hand, an obvious choice because of its implication in LTP. However, evidence in the literature suggests that LTP in the somatosensory cortex can be GluA1-independent (e.g., Frey et al., 2009). The authors should discuss these studies' implications and the limits of their method for addressing synaptic mechanisms for learning and memory. Does SEP-GluA1 fluorescence reproduce previous estimates of the distribution of synaptic weights? The SEP-GluA1 method is undoubtedly more attractive than the post hoc patch-clamp-based approaches performed by Ko et al. 2011, 2013 (Hofer and Mrsic-Flogel laboratories).

6) Conditional expression of the SEP-GluA1 would allow the precise dissection of synaptic plasticity specific neuron types within microcircuits, thereby linking plasticity with altered circuit function and behavior. This limitation must be addressed, in detail, in the Discussion. The current data sets of whole-brain expression of SEP-GluA1 do not distinguish between layer 2/3 pyramidal neurons, layer 5 pyramidal neurons, or interneurons. It is interesting to consider the possibility that a cross between the SEP-GluA1 transgenic mouse and a PSD-95 YPF-Enabled (Fortin et al., 2014) mouse could be used to assess the GluA1 changes at specific synapse types.

7) Using Western Blot, the authors show a 57% reduction of GluA1 expression at the postsynaptic density relative to WT, however, the potential implications of this are not addressed. In figure 1f, they show relative AMPA levels resulting from their Western blot analysis. Still, they do not adequately describe how the Western blot and analysis were performed, and namely, what control they used to normalize protein levels between different fractions. It would be nice to see additional statistical analyses of GluA2 and GluA3 expression relative to WT as there seems to be an increase and, therefore, potential compensation by GluA3 subunits SEP-GluA1 KI.

8) In the FRAP experiment in Figure 6, it seems like there could be other interpretations for the time course of recovery of fluorescence. The expected timescales of fluorescence recovery and caveats of its interpretations should be discussed.

[Editors' note: further revisions were suggested prior to acceptance, as described below.]

Thank you for resubmitting your work entitled "Visualizing synaptic plasticity in vivo by large-scale imaging of endogenous AMPA receptors" for further consideration by *eLife*. Your revised article has been reviewed by 3 reviewers, one of whom is a member of our Board of Reviewing Editors, and the evaluation has been overseen by Lu Chen as the Senior Editor.

The manuscript has been improved but there are some remaining issues that need to be addressed, as outlined below:

1. There is still no convincing data that spines can be tracked in vivo across a long time interval (e.g. 1 day). The authors show a picture with sparse Td-tomato labelling, but only on a single time point. It is necessary to show the same picture at a later time point to appreciate the degree of consistency of SEP-GluA1 labelling across days. The sentence in the abstract, "we can longitudinally track how the strength of synapses changes during behavior," is currently not fully supported by the data.

2. The authors did not compare the efficacy of their automated synapse segmentation methods to other methods. The authors state in the discussion that their method outperforms other methods but this is not documented. This statement requires a quantification.

3. The authors do not convincingly demonstrate that changes in synaptic strength can be quantified by relative changes in GluR1 fluorescence since the fraction of GluR1 containing receptors is different at rest and after plasticity, thus overestimating changes in plasticity. Perhaps Figure 5d and e can be better leveraged to show similar relative changes in EPSC and SEP-GluA1. Adequate statistics should be shown for this point.

4. The following sentences in the abstract should be changed:

• "We used this approach to generate an unprecedentedly detailed spatiotemporal map of synaptic plasticity underlying sensory experience" Change this to "a map of synapses undergoing changes in synaptic strength."

• "More generally, these tools can be used as an optical probe capable of measuring functional synapse strength" This should be changed to identifying synapses undergoing increased synaptic strength.

5. How do the authors estimate the error in synaptic strength estimates contributed by multiple synapses within the extended PSF volume? It is unclear how to differentiate between multiple synapses and one synapse with a high GluA1 content. The PSF resolution is particularly concerning for the high density of synapses observed in the cortex in vivo. The author should state this issue and recommend post hoc immunocytochemical labeling of synaptic markers (e.g., Homer or PSD95).

6. The following statement (page 15) is subjective and should be quantified: "Presumably, this trade-off led to increased false-negative rates, but reduced false-positive rates that would have arisen from incorrectly merging two distinct synapses."

7. Was the PSF performed with 1 μm beads? It seems impossible to have measured a FWHM of 0.57 um.

---

## [Author Response]

Essential revisions:1) The authors should clarify how the synapse detection algorithm account for extrasynaptic GluA1 labeling. Increases in this pool may not directly underlie increases at this synapse. Testing these confounding factors could be performed using physiologically relevant plasticity protocols in acute brain slices rather than cell culture, a preparation in which the neuron type and connectivity may not be the same as the mostly intact circuit.

In neurons, membrane-inserted AMPA receptors are mostly clustered at functional postsynaptic sites on dendritic spines or the dendritic shaft. It is true, however, that a minority population of extrasynaptic AMPA receptors are present along the plasma membrane outside of these synaptic zones. To reduce false-positive detections of extrasynaptic SEP-GluA1 as functional synapses, we employed an automated template-matching approach that was trained on manually annotated synaptic puncta. Detections were filtered based on size in both the XY and Z planes (see description in the methods section). These size exclusion thresholds were chosen based on the known size of AMPA-containing PSDs from EM. We found that this approach performed well in excluding the extrasynaptic SEP-GluA1 signal, which tended to be smaller, more diffuse, and less punctate than synaptically enriched GluA1. We thus believe that our analysis of synaptic SEP-GluA1 levels only has minimal contributions of extrasynaptic GluA1. Nevertheless, this is a crucial point and that we have more clearly emphasized in the manuscript (page 13), detailing how our algorithm was optimized to detect synaptic GluA1 and largely exclude extrasynaptic signals.

2) The extended z-resolution of the 2P point-spread-function (PSF) in vivo will likely falsely assign fluorescence from multiple synapses to a single punctum. How does this limited resolution impact the final interpretations? The study would benefit from an in vivo ground-truth estimate of synapse location to confirm that SEP-GluA1 puncta correspond to an increase in AMPARs at synapses. The use of Homer does not entirely circumvent the uncertainty generated by the poor resolution of 2P. Post-hoc confocal microscopy in fixed tissue using morphology and synaptic markers could be sufficient. If not, then super-resolution fluorescence or immunogold EM are certainly suitable ground-truth approaches.

We agree with the reviewers that due to inherent constraints of in vivo light microscopy, our ability to accurately detect and segment closely neighboring synapses is somewhat limited. This is especially true in the z-plane, where the point spread function (PSF) of 2p imaging limits resolution and detection, far more so than in the XY planes. Throughout our manuscript, we acknowledge that while our endogenous knockin strategy does fluorescently label all GluA1 subunits, we are not able to detect every single GluA1-containing synapse. To quantitatively assess the limitations of in vivo 2p imaging and subsequent synapse detection, we have included measurements of our 2P PSF (see new Figure 7 Supplement 3). We have updated the manuscript to explicitly state these resolution constraints (page 15), as well as the conclusion that we cannot accurately detect and segment synapses that are within these bounds (i.e., that we cannot accurately discern and segment two overlapping synapses in XY that are within 3 μm in Z). We have expanded discussion regarding how these limitations impact our analyses (page 15), since our algorithm would detect and segment these closely abutting synapses as one large synapse, and ultimately discard this “false positive” synapse for being too big (detections that span more than 6 adjacent z-planes are discarded).

3) The quantification of synaptic strength by SEP-GluA1 may be an overestimate because SEP-GluA1 containing AMARs represents a fraction of the total receptor population, and LTP is strongly biased towards GluA1 insertion. How can this potential error be overcome?

We believe that the abundance of GluA1 serves as a valuable proxy for synaptic strength and demonstrate this empirically with the correlation between SEP-GluA1 signal and EPSC amplitude with glutamate uncaging (Figure 5). Regarding a potentially privileged role of GluA1 in LTP, this is why we sought to focus on GluA1 in particular, to aid the identification of synapses where there are plasticity changes. While this lies outside of the scope of this study, we are also working on developing SEP-KI mouse lines for other AMPAR subunits which will ultimately allow us to study differences in subunit specific AMPAR plasticity and how each subunit contributes to changes in the total receptor population. However, we accept the reviewers’ concerns and have modified our discussion (page 17) to highlight that SEP-GluA1 changes might indicate synapses preferentially undergoing plasticity.

4) To verify that SEP-GluA1 can reliably report decreases in synaptic strength, the authors should examine the activity-dependent endocytosis of receptors following LTD pairing protocols, preferably from the same cell type in the brain slices.

AMPARs, including GluA1-containing receptors, mediate both increases and decreases in synapse strength, corresponding to LTP and LTD, respectively (Malenka and Nicoll, Science 1999; Carroll et al., Nat Neuroscience 1999; Shepherd and Huganir, Annu Rev Cell Dev Biol 2007; Huganir and Nicoll, Neuron 2013). We agree with the reviewers that it is important that our SEP-GluA1 mouse can track both increases and decreases in synaptic GluA1 content, since it is well-established that synaptic weakening, such as during LTD or homeostatic down-scaling, depends on removal of synaptic AMPARs. However, we do not believe there is reason to expect that our SEP-GluA1 reporter is not bidirectional. Indeed, in our manuscript, we include biochemical data directly demonstrating that SEP-GluA1 is capable of reporting both increased and decreased AMPAR expression (see homeostatic plasticity experiments in Figure 3a-b). We have expanded discussion of this crucial point (page 19). Additionally, we have included longitudinal imaging data from individual spines from our FRAP experiments which show that in the unbleached controls spines that our SEP-GluA1 KI line can report increases and decreases in spine GluA1 levels at individual synapses across imaging sessions (new Figure 6c). These results are additionally supported by previous studies where bidirectional changes in spine GluA1 levels were observed while imaging sparsely expressed SEP-GluA1 in vivo, (Roth et al., 2020; Tan et al., 2020). We performed the slice LTP experiments (Figure 3), as well as the 2p uncaging experiments (Figure 5) as a control, to demonstrate that Hebbian plasticity is not impaired in our knockin and to directly demonstrate that SEP intensity directly correlates with functional synaptic strength. We thus believe that the proposed slice LTD experiments would not add anything further to demonstrate the capacity of SEP-GluA1 to report bidirectional changes in synaptic AMPAR content.

5) The choice to fluorescently label GluA1 subunits of the AMPAR to mark plasticity is, on the one hand, an obvious choice because of its implication in LTP. However, evidence in the literature suggests that LTP in the somatosensory cortex can be GluA1-independent (e.g., Frey et al., 2009). The authors should discuss these studies' implications and the limits of their method for addressing synaptic mechanisms for learning and memory. Does SEP-GluA1 fluorescence reproduce previous estimates of the distribution of synaptic weights? The SEP-GluA1 method is undoubtedly more attractive than the post hoc patch-clamp-based approaches performed by Ko et al. 2011, 2013 (Hofer and Mrsic-Flogel laboratories), but SEP-GluA1 should be compared to spine intensity (volume) imaging for example.

We agree with the reviewer that our SEP-GluA1 KI mouse line can only track plasticity changes involving GluA1. We have now modified our discussion to highlight that SEP-GluA1 changes might indicate synapses preferentially undergoing plasticity and that different types of plasticity might involve GluA1 to a different degree (page 20). Throughout our manuscript, we are clear that our SEP-GluA1 knockin line enables tracking of GluA1 expression and not other AMPAR subunits. This is not to say that only GluA1 expression is changing during the behaviors and experimental manipulations in this study, nor that GluA1-mediated synaptic plasticity is the sole mechanism of behavioral learning, in the sensorimotor cortex or anywhere else in the brain. Indeed, different forms of plasticity can differentially rely on different AMPAR subunits and we are developing SEP-tagged mouse lines for other AMPAR subunits that can be used to distinguish these in the future. However, as mentioned above in Point 3, we believe that the abundance of GluA1 serves as a valuable proxy for synaptic strength and we demonstrate this here with the correlation between SEP-GluA1 signal and EPSC amplitude with glutamate uncaging (Figure 5). Frey et al. (2009) show that LTP in somatosensory cortex can be induced in the absence of GluA1, however this does not mean that GluA1 levels are not increased during LTP in WT mice. Our lab’s previous work has demonstrated that whisker stimulation can indeed induce NMDA-receptor-dependent LTP that involves insertion of GluA1 to dendritic spines in vivo (Zhang et al., 2015).

Regarding the comparison of SEP-GluA1 and spine volume, we believe that tracking SEP-GluA1 is a more accurate readout of synaptic strength than spine volume since the strength is mediated by receptors rather than spine size. Indeed, while spine size generally correlates with SEP-GluA1 intensity, spine size does not always change during plasticity (Figure 3A and B in Zhang et al., 2015) or changes to a lesser degree than SEP-GluA1 (Tan et al., 2020, Roth et al., 2020). We even observed a population of spines that exhibit a divergence of spine form and function, with spine SEP-GluA1 increasing but spine size decreasing and vice versa (Tan et al., 2020). The dissociation of spine size and synaptic strength has been reported many times (Lee et al., 2012). For instance, spine number or volume is not changed at all at cerebellar Purkinje cell synapses during LTD (Sdrulla and Linden, 2007) and Insulin-induced endocytosis of AMPARs is not accompanied by spine shrinkage (Wang et al., 2007). Thus, spine size, in certain conditions, is not a good indication of synaptic strength, whereas our imaging of synaptic AMPAR expression provides a direct and accurate way to monitor functional changes at synapses.

6) Conditional expression of the SEP-GluA1 would allow the precise dissection of synaptic plasticity specific neuron types within microcircuits, thereby linking plasticity with altered circuit function and behavior. This limitation must be addressed, in detail, in the Discussion. The current data sets of whole-brain expression of SEP-GluA1 do not distinguish between layer 2/3 pyramidal neurons, layer 5 pyramidal neurons, or interneurons. It is interesting to consider the possibility that a cross between the SEP-GluA1 transgenic mouse and a PSD-95 YPF-Enabled (Fortin et al., 2014) mouse could be used to assess the GluA1 changes at specific synapse types.

We agree that a Cre-conditional SEP-GluA1 line – and, for that matter, conditional versions of all AMPAR subunit lines – would be valuable tools to have and would enable important cell-type-specific investigations into synaptic plasticity in vivo. This is the first paper demonstrating that a SEP knockin mouse line can be used to track synaptic AMPARs in general. We are currently trying to generate these conditional lines and future studies by our lab and others will continue to advance this line of inquiry. However, designing a conditional knockin at the N-terminus of an endogenous protein is highly challenging. While it might be hard to separate YFP fluorescence of the PSD-95 ENABLED mouse from our SEP-GluA1 fluorescence, the idea of combining a conditionally expressed cell or synaptic marker in a different color is a possible approach to achieve cell-type specificity in our global SEP-GluA1. Such a marker could be a red PSD-95 intrabody (Gross et al., 2013) or a tdTomato cell fill. In this study we have combined the SEP-GluA1 mouse with the Ai9 tdTomato reporter mouse and injected AAV-CaMKII-cre virus in L2/3 pyramidal neurons. This approach allowed us to specifically associate SEP-GluA1 synapses to dendrites of cre expressing neurons (Figure 7 c-d) and target these for FRAP (Figure 6). These results demonstrate that it is feasible to combine our SEP-GluA1 knockin line with a mouse line expressing tdTomato in the cell-type of interest and specifically quantify changes in spine GluA1 expression of these neurons. We have expanded our discussion and emphasized that combining our global SEP-GluA1 knockin with conditionally expressed red cell or synaptic markers is a viable approach to achieve cell-type specificity (page 21).

7) Using Western Blot, the authors show a 57% reduction of GluA1 expression at the postsynaptic density relative to WT, however, the potential implications of this are not addressed. In figure 1f, they show relative AMPA levels resulting from their Western blot analysis. Still, they do not adequately describe how the Western blot and analysis were performed, and namely, what control they used to normalize protein levels between different fractions. It would be nice to see additional statistical analyses of GluA2 and GluA3 expression relative to WT as there seems to be an increase and, therefore, potential compensation by GluA3 subunits SEP-GluA1 KI.

We do not know why the mRNA and GluA1 is lower than WT in these mice. It is likely to be due to RNA instability of the KI transcript. We will make attempts in the future to limit this issue. However, this decrease in GluA1 expression does not affect synapse function and plasticity. We are sorry we did not describe the methods and quantitation of the western blots in more detail. Equal amounts of protein were run in each lane and PSD95 was used as internal control and GluA1 levels were normalized to PSD95 for both P2 and PSD analysis. There was no significant difference in PSD95 level between WT and SEP-GluA1 KI. In addition, we used the raw data without any normalization with PSD95 for analysis and observed similar changes: 43.45% reduction in GluA1 after normalization to PSD95 v.s. 43.31% reduction with normalization to the amount of protein loaded (PSD fraction, Author response image 1). Regarding GluA2 and GluA3 expression, we saw a trend of increase (15.77% increase in GluA2 and 25.61% increase in GluA3 at PSD after normalization to PSD95) although the increases were not statistically significant (Author response image 1) . It is possible that there is a compensatory increase in GluA2/GluA3 subunits in SEP-GluA1 KI animals, however, our statistical analysis does not support this claim. We have included the new statistical analysis in the revised manuscript and updated our discussion of these results accordingly (page 6).

**Author response image 1. sa2fig1:** Quantification of AMPA receptor subunit expression with or without normalization to PSD95 in the P2 and postsynaptic density (PSD) fractions of WT and SEP-GluA1 mice. (n=7; *p < 0.05; ****p < 0.0001, Student’s T-test).

8) In the FRAP experiment in Figure 6, it seems like there could be other interpretations for the time course of recovery of fluorescence. The expected timescales of fluorescence recovery and caveats of its interpretations should be discussed.

We have clarified our interpretation of our FRAP experiments and expanded our discussion about the time course of fluorescence recovery (page 11). Here is the new text:

“We also found that the SEP-GluA1 signal recovered in two phases after photobleaching (Figure 6b). […] Overall, these results confirm that SEP-GluA1-containing synapses are mobile and present similar dynamics as other in vitro and in vivo systems, supporting that our knockin labelling strategy does not perturb normal synaptic dynamics or function.”

9) An essential aspect of plasticity in the brain is that it can occur on long time scale on precise synapses. Here the authors do not show that it is possible to do longitudinal studies in which individual synapses are tracked. The authors need to provide some data to allow evaluating if this this is actually possible with their method. e.g. with a cytoplasmic marker for spines.

We agree with the reviewers regarding the importance of tracking individual synapses over time to understand mechanisms of synaptic plasticity. To address this point, we have included additional imaging data that shows the capacity of our approach to track longitudinal changes in spine GluA1 levels at individual synapses across multiple days (new Figure 6c). Moreover, representative images of individual spines across imaging sessions are shown in figure 6a. In these experiments, a tdTomato cell fill was expressed in a sparse population of neurons to facilitate identification of the same dendritic spines at different time points. This demonstrates that, at least with the addition of a cytoplasmic marker as the reviewer suggests, it is possible to track individual spines across imaging sessions that span days. Our updated discussion of these new data appears on page 11.

10) Comparison of their synapse detection algorithm with concurrent methods is necessary to establish it as a new tool.

We have greatly expanded discussion of our automatic synapse detection algorithm, both in terms of how it was constructed (explicitly listing the rules for synapse detection, on page 13) and how it is differentiated from other similar tools (page 22). Here is the new text:

“Recently, deep-learning-based systems have achieved state-of-the-art performance in analyzing microscopy images (Moen et al., 2019). […] We found that none of these approaches performed as well as our rules-based, template-matching approach to automatically detect SEP-GluA1-containing synapses.”

Reviewer #1 (Recommendations for the authors):1) Better discuss the limits of synapse identification on images. The limits are related both to the algorithm used and to 2-phoron imaging resolution limits. Thus better identification could be obtained maybe with super resolution methods. Ideally providing ground truth data would be an asset.

We agree with the reviewer that accuracy of our synapse identification is largely limited by the resolution of two-photon microscopy, and by extension how our automatic synapse detection algorithm is similarly limited by these fundamental constraints. We have now quantified the limits of our synapse detection by measuring the point spread function (PSF) of our microscope and discuss the parameters of our detection algorithm in more detail. We further agree with the reviewer that in vivo super resolution imaging would improve synapse detection, especially in the X and Y dimension, and can be used as an approach in experiments where a more detailed look at individual synapses and subsynaptic structures is necessary. For further details please see our response to this point under “Essential Revisions Point 2”.

2) An essential aspect of plasticity in the brain is that it can occur on long time scale on precise synapses. Here the authors do not show that it is possible to do longitudinal studies in which individual synapses are tracked. I recommend to show some data that it is actually possible with their method, e.g. with a cytoplasmic marker for spines.

Please see our response to this point under “Essential Revisions Point 9”.

3) Comparison with previous methods (e.g. spine volume imaging) is necessary to establish this tool as a useful to track synaptic plasticity. This could be done in the whisker-based plasticity assay.

Please see our response to this point under “Essential Revisions Point 5”.

Reviewer #2 (Recommendations for the authors):This is an important paper with well carried-out experiments. The conclusions are well supported by data. I have however 3 concerns (that do not dampen my enthusiasm):1) The global expression of GluA2 is twice as low as the expression of GluA1 in wild-type mice, although the level of GluA2 remains unchanged. How would this affect GluA1/Glu2 heteromers? In addition, can the authors exclude the possibility that the decrease of SEP-GluA1 is brain region specific? (e.g. different levels of SEP-GluA1 between the hippocampus and the cortex). if the decrease is indeed region-specific, could the author exclude that physiological deficit does not occur in other brain region? At best, the authors should perform a better comparison of GluA1 and SEP-GluA1 between brain regions. A least, these point should be discussed. In the same line, bidirectional changes after Bic and TTX treatment in culture are really convincing (Figure 3B). However, it would be fair to present also the non-normalized level of surface AMPAR after treatment as compared to WT.

We agree with the reviewer that it is important to understand region-specific levels of SEP-GluA1 expression in our knockin mouse and how the reduction in GluA1 can affect AMPAR subunit compositions. Our Western blot data unfortunately cannot distinguish GluA1/GluA2 heteromers from GluA1 homomers. Indeed, it is technically challenging to specifically examine GluA1/GluA2 heteromers. In our experiments, we saw a trend of an increase in GluA2 and GluA3 levels at PSD of SEP-GluA1 KI mice compared with WT mice although the increases were not statistically significant (see our response to this point under “Essential Revisions Point 7”). Therefore, it is possible that there is a reduction in GluA1/GluA2 heteromers accompanied by an elevation of GluA2/GluA3 heteromers in SEP-GluA1 KI mice.

We have included new comparisons of SEP-GluA expression to patterns of GluA1 mRNA level in WT mice. These atlases compare fluorescent SEP-GluA1 expression throughout the brain, with aligned references from Allen Brain Atlas (new Figure 1 Supplement 1).

Regarding the region-specific levels of SEP-GluA1 expression, we don’t believe that there are any differences between brain regions. We observed a reduction in GluA1 level in hippocampus (Figure 1e-f). We observed a similar reduction in surface SEP-GluA1 level in cultured cortical neurons of KI pups compared to WT (Figure 3a). Additionally, we performed new experiments using Western blots from either whole brain or specific brain regions and observed a similar pattern of decrease of total GluA1 in KI mice, irrespective of brain region (new Figure 1 Supplement 2). Therefore, we believe that the expression level of GluA1 in SEP-GluA1 mice is conserved across brain regions.

Regarding Figure 3b, we think it is more visually representative of the similarity of the homeostatic plasticity between WT and the KI by normalizing the data.

2) Although the authors present strong and convincing evidence that the high density of SEP-GluA1 synapses does not preclude the report of synaptic plasticity in anesthetized mice, it is much less clear whether it would work also in awake mice.

Our previous studies have shown that we can longitudinally image dendritic/spine SEP-GluA1 to study AMPAR dynamics both in anesthetized and head-fixed awake mice during different kinds of behaviors with the same set-up (Zhang et al., 2015; Roth et al., 2020; Tan et al., 2020). Our ability to resolve and track individual dendritic spines was identical between these two groups. Therefore, we are confident that imaging awake SEP-GluA1 mice is no different from imaging anesthetized mice.

3) How effective is their supervised detecting method as compared to existing computer-based detection methods ? (e.g. see figure 7 from: https://www.biorxiv.org/content/10.1101/061507v2.full.pdf). Also, how robust is the supervised annotation of spines when the same FOV is imaged chronically (between days)?

We agree with the reviewer that chronic imaging of the same FOV across days is essential for understanding synaptic plasticity during learning. While currently our machine-learning-based detection algorithm is focused on detecting synapses on a single imaging session, in future studies, we will expand on our approach to align, register, and track millions of individual synapses across days. Currently, we can achieve reliable longitudinal imaging by including cytosolic tdTomato in a sparse population of neurons to facilitate the identification of the same dendritic spines at the different time points. We have now included additional imaging data that shows the capacity of our approach to track longitudinal changes in spine GluA1 levels at individual synapses across multiple imaging sessions (Author response image 1 and new Figure 6c). Moreover, representative images of individual spines across imaging sessions are shown in figure 6a. This demonstrates that, at least with the addition of a cytoplasmic marker, it is possible to track individual spines across imaging sessions that span days. As suggested by the reviewer, we have also greatly expanded discussion of our automatic synapse detection algorithm, both in terms of how it was constructed and how it is differentiated from other similar tools. Please see our response to this point under “Essential Revisions Point 10”.

[Editors' note: further revisions were suggested prior to acceptance, as described below.]

The manuscript has been improved but there are some remaining issues that need to be addressed, as outlined below:1. There is still no convincing data that spines can be tracked in vivo across a long time interval (e.g. 1 day). The authors show a picture with sparse Td-tomato labelling, but only on a single time point. It is necessary to show the same picture at a later time point to appreciate the degree of consistency of SEP-GluA1 labelling across days. The sentence in the abstract, "we can longitudinally track how the strength of synapses changes during behavior," is currently not fully supported by the data.

We agree with the reviewer that our data do not support a claim that we are able to track individual, registered synapses over time. Rather, our data clearly support that we can track how populations of synapses change during sensory stimulation. While we are able to detect hundreds of thousands of individual synapses, longitudinal registration and tracking of each of these synapses is beyond the scope of this study. We have updated the abstract, results, and discussion to explain our approach and the efficacy of our tools more clearly.

2. The authors did not compare the efficacy of their automated synapse segmentation methods to other methods. The authors state in the discussion that their method outperforms other methods but this is not documented. This statement requires a quantification.

We have updated our discussion of other automated synapse detections methods, removing any claims that our tools outperform other approaches. The reviewers are correct that we do not have the data to make this claim.

3. The authors do not convincingly demonstrate that changes in synaptic strength can be quantified by relative changes in GluR1 fluorescence since the fraction of GluR1 containing receptors is different at rest and after plasticity, thus overestimating changes in plasticity. Perhaps Figure 5d and e can be better leveraged to show similar relative changes in EPSC and SEP-GluA1. Adequate statistics should be shown for this point.

It is not clear why this reviewer states that the fraction of GluA1 containing receptors change during plasticity. This concept is not accepted by many people in the field and has certainly has not been shown in vivo. Our data support that: (1) SEP-GluA1 intensity is strongly correlated with functional synaptic strength in vitro (Figure 5c), (2) LTP in vitro is associated with increased SEP-GluA1 intensity only in spines that received the plasticity-inducing stimulus (Figure 5d-f), and (3) that SEP-GluA1 intensity and EPSC amplitude are also strongly correlated after plasticity (Figure 5 Supp. 1). To more fully support this final point, we include new plots showing this correlation after LTP in Figure 5 Supplement 1. We feel that these data and statistical analyses fully address this point.

4. the following sentences in the abstract should be changed:• "We used this approach to generate an unprecedentedly detailed spatiotemporal map of synaptic plasticity underlying sensory experience" Change this to "a map of synapses undergoing changes in synaptic strength."• "More generally, these tools can be used as an optical probe capable of measuring functional synapse strength" This should be changed to identifying synapses undergoing increased synaptic strength.

We have changed the wording of the abstract based on the first point as suggested. However, we disagree with the second point. Firstly, as we said in the first round of reviewer comments (Essential Revision #4), our methods allow us to directly observe both increases and decreases in synaptic strength in vivo, not just increased synaptic strength, as this comment suggests. Indeed, we present data that demonstrate we can track both increases and decreases in synaptic strength over time (Figure 6). Further, given our glutamate uncaging data in Figure 5c, our results strongly support the claim that in vivo imaging of this SEP-GluA1 mouse “can be used as an optical probe capable of measuring functional synapse strength”. Thus, we feel strongly that this sentence should remain, unchanged.

5. How do the authors estimate the error in synaptic strength estimates contributed by multiple synapses within the extended PSF volume? It is unclear how to differentiate between multiple synapses and one synapse with a high GluA1 content. The PSF resolution is particularly concerning for the high density of synapses observed in the cortex in vivo. The author should state this issue and recommend post hoc immunocytochemical labeling of synaptic markers (e.g., Homer or PSD95).

We discuss this issue and describe our rules for defining synapses in a numbered list are clearly stated on page 13. Fluorescent puncta that obey all these rules are considered synapses. These parameters were carefully chosen, based on published ranges of synapse size, as well as our experience imaging labeled synaptic proteins in vitro and in vivo. The reviewer is correct that inherent limitations of light microscopy—especially the comparatively broad axial PSF—complicates detection and segmentation of densely labeled synapses, and could theoretically lead to cases where two closely abutting synapses are erroneously detected and segmented as one large synapse. To minimize this confound, we excluded synapses that spanned more than 6 adjacent z-sections (rule #8). We state this in the text (pages 13 and 16).

Based on these comments, we have also substantially expanded (pages 16-17) discussion of the potential pitfalls of detecting densely labeled synapses, and how we think our data support that we’re not making a significant number of merge errors. Briefly, if we were making a significant number of merge errors – erroneously merging two or more closely abutting synapses into a single automatic detection – the we should see additional peaks in histograms of synapse intensity (e.g., peaks at 2x and 3x of max intensity, arising from detections containing 2 or 3 synapses). We don’t see any evidence of this in our data. Rather, distributions of automatically detected synapses are strikingly normal, supporting that an overwhelming majority of our detected synapses correspond to single synapses, despite their high density and the poor axial PSF inherent to 2p imaging. However, we have added a statement in the updated manuscript that in future studies using post hoc immunocytochemical labeling of synaptic markers (e.g., Homer or PSD95) could be useful.

6. The following statement (page 15) is subjective and should be quantified: "Presumably, this trade-off led to increased false-negative rates, but reduced false-positive rates that would have arisen from incorrectly merging two distinct synapses."

We have updated the manuscript to more clearly state our rationale for excluding two (or more) closely opposed synapses that could not be adequately segmented. We also removed the subjectively worded sentence flagged by the reviewers. Many of these points are addressed in Point #5, above.

7. Was the PSF performed with 1 μm beads? It seems impossible to have measured a FWHM of 0.57 um.

The PSF we present is a "Reconstructed PSF from averaged two-photon images of 1μm TetraSpeck microspheres", which is computed through an inverse deconvolution of raw images following established methods (Padman, 2014; Schmied et al. 2016; Verveer et al. 2020). This process is necessary to extract the PSF from finite (non-zero) sized beads, as using the raw bead images directly will give an overestimation of the PSF. This point is clarified in the Results section of the updated manuscript (pages 15-16).